# The plasmidome associated with Gram-negative bloodstream infections: A large-scale observational study using complete plasmid assemblies

Samuel Lipworth [1,2,7] ✉, William Matlock [1,7], Liam Shaw [3], Karina-Doris Vihta [4], Gillian Rodger[1], Kevin Chau[1], Leanne Barker[1], Sophie George[1], James Kavanagh [1], Timothy Davies[1,3], Alison Vaughan[1], Monique Andersson [2], Katie Jeffery [2], Sarah Oakley[2], Marcus Morgan[2], Susan Hopkins[5], Timothy Peto[1,2,6], Derrick Crook[1,2,6], A. Sarah Walker[1,6,8] & Nicole Stoesser [1,2,6,8]

Plasmids carry genes conferring antimicrobial resistance and other clinically important traits, and contribute to the rapid dissemination of such genes. Previous studies using complete plasmid assemblies, which are essential for reliable inference, have been small and/or limited to plasmids carrying antimicrobial resistance genes (ARGs). In this study, we sequenced 1,880 complete plasmids from 738 isolates from bloodstream infections in Oxfordshire, UK. The bacteria had been originally isolated in 2009 (194 isolates) and 2018 (368 isolates), plus a stratified selection from intervening years (176 isolates). We demonstrate that plasmids are largely, but not entirely, constrained to a single host species, although there is substantial overlap between species of plasmid gene-repertoire. Most ARGs are carried by a relatively small number of plasmid groups with biological features that are predictable. Plasmids carrying ARGs (including those encoding carbapenemases) share a putative 'backbone' of core genes with those carrying no such genes. These findings suggest that future surveillance should, in addition to tracking plasmids currently associated with clinically important genes, focus on identifying and monitoring the dissemination of high-risk plasmid groups with the potential to rapidly acquire and disseminate these genes.

Gram-negative bloodstream infections (BSI) are associated with substantial morbidity and mortality; their incidence continues to increase both in the UK and globally[1,2]. Multi-drug-resistant and hypervirulent phenotypes are a particular concern, especially since genes conferring these characteristics (and others which may have either positive or negative fitness effects) are carried on plasmids, frequently in association with other smaller mobile genetic elements[3,4]. Plasmids are thought to facilitate the rapid dissemination of these genes within and

[1]Nuffield Department of Medicine, University of Oxford, Oxford, UK. [2]Oxford University Hospitals NHS Foundation Trust, Oxford, UK. [3]Department of Zoology, University of Oxford, South Parks Road, Oxford, UK. [4]Department of Engineering Science, University of Oxford, Oxford, UK. [5]National Infection Service, United Kingdom Health Security Agency, Colindale, London, UK. [6]NIHR Oxford Biomedical Research Centre, John Radcliffe Hospital, Oxford, UK. [7]These authors contributed equally: Samuel Lipworth, William Matlock. [8]These authors jointly supervised this work: A. Sarah Walker, Nicole Stoesser. ✉e-mail: samuel.lipworth@ndm.ox.ac.uk

between bacterial species. A detailed understanding of their biology and epidemiology is therefore likely to be crucial in tackling the global threat of antimicrobial resistance (AMR).

Complete and accurate genome assemblies, such as those produced by "hybrid" assemblies of short and long-read sequencing data, are crucial for the study of plasmid epidemiology. Until recently, however, these have been prohibitively expensive for large-scale application, and so whilst this approach has recently been used at scale to evaluate the plasmidome of environmental/agricultural isolates[5], to our knowledge its application to human-associated isolates has been mostly restricted to relatively small numbers of isolates selected based on AMR phenotype[3,6]. This phenotype-driven selection strategy has identified several plasmid types associated with the dissemination of key ARGs but it is currently not known whether similar plasmids are also found in susceptible populations. Earlier works have demonstrated the similarity of ARG-associated plasmids from the pre- and post-antibiotic era[7–9], hinting that these genes are disseminated on amongst well-conserved pre-existing plasmid families. Recently, two studies have demonstrated the utility of network-based approaches to classify plasmid assemblies from public databases, offering insights into the host range of these plasmids, though such analyses suffer from sampling bias as well as a lack of clinical context and accurate metadata[10,11]. Therefore, the plasmidome associated with Gram-negative isolates causing both antimicrobial susceptible and sensitive clinical infections remains largely uncharacterised.

Using short-read sequencing we have previously described in detail the population dynamics of *E. coli* and *Klebsiella* spp. BSI isolates collected between 2009 and 2018 in Oxfordshire, UK[12]. In this study using hybrid assembly, we generated complete genomes for all isolates collected in 2009 and 2018, as well as a representative sample from

intervening years. Using this dataset, we first sought to investigate the extent to which plasmids are shared and contribute to overlaps in the pangenome within and between species. We then sought to compare plasmids associated with ARG carriage to those that are not. Subsequently, we investigated the dissemination dynamics of the most prevalent ESBL gene in the population, $bla_{CTX-M-15}$, highlighting complex nested mobilisation that can only be unravelled using hybrid assembly. Finally, we contextualised our findings by comparing our plasmid dataset to a large global collection and investigated whether features of "successful" plasmids and those with the potential for ARG carriage are predictable.

## Results

We sequenced and assembled $n = 953$ isolates of which $n = 738$ were complete and included in subsequent analysis (Supplementary Fig. S1). Of these, 75% (553/738) were *E. coli* ($n = 153, 297, 103$ in 2009, 2018, intervening years, respectively), 22% (161/738) *Klebsiella* spp. ($n = 39, 58, 64$ in 2009, 2018, intervening years, respectively) and 3% (24/738) other Enterobacterales species (details in Supplementary Fig. S1 and Supplementary Data 3). In total, these 738 isolates carried 1,880 plasmids with a median of 2 plasmids per isolate (interquartile range (IQR) 1–3). 10% (77/738) isolates carried none, 29% (211/738) carried one and 61% (450/738) more than one (Fig. 1A). Of the $n = 661/738$ isolates with at least one plasmid, 77% (508/661) carried at least one large plasmid (i.e., sequence length >100,000 bp), and 94% 621/661) at least one large or medium plasmid (i.e., sequence length >10,000 bp); of these 53% (329/621) also carried at least one small plasmid (i.e., sequence length <10,000 bp). Carriage of one or more small plasmids in the absence of any medium or large plasmid was relatively rare at 6% (40/661). Rarefaction analysis suggested that a substantial number of

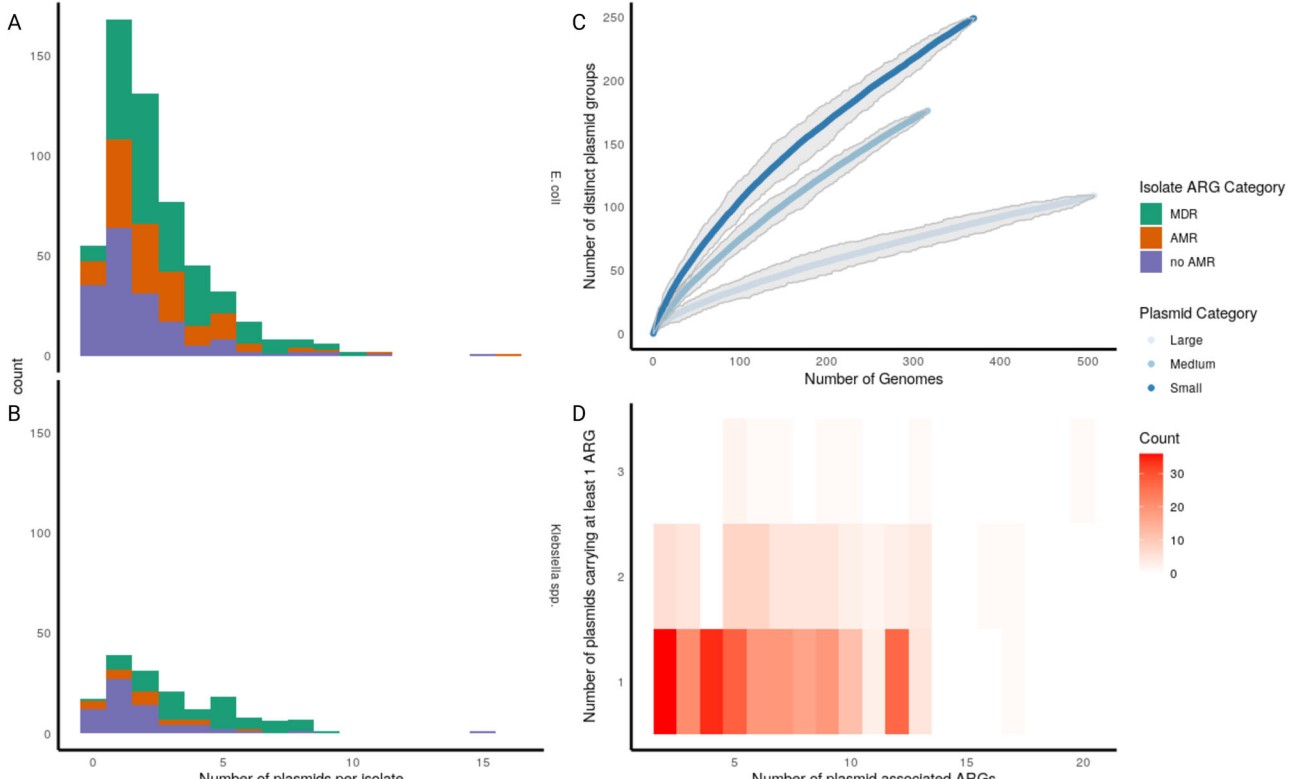

**Fig. 1 | Characteristics of plasmid carriage in *E. coli*/*K pneumoniae* bloodstream infections. A**, **B** Number of plasmids per isolate for *E. coli* (**A**) and *Klebsiella* spp. (**B**), coloured by the number of ARG classes per isolates where MDR is ≥3, AMR 1–2 and no AMR 0. **C** Rarefaction curve of the number of novel plasmid groups (as defined using the Louvain-based method described above) per new plasmid sequenced stratified by size (large ≥100,000 bp, medium ≥10,000 to < 100,000 bp, small < 10,000 bp. **D** Number of plasmid-associated ARGs per isolate vs number of plasmids carrying at least one ARG. Isolates with only one plasmid-associated ARG (by definition, carried on one plasmid) are excluded. Source data are provided in the supplementary "Source Data" file.

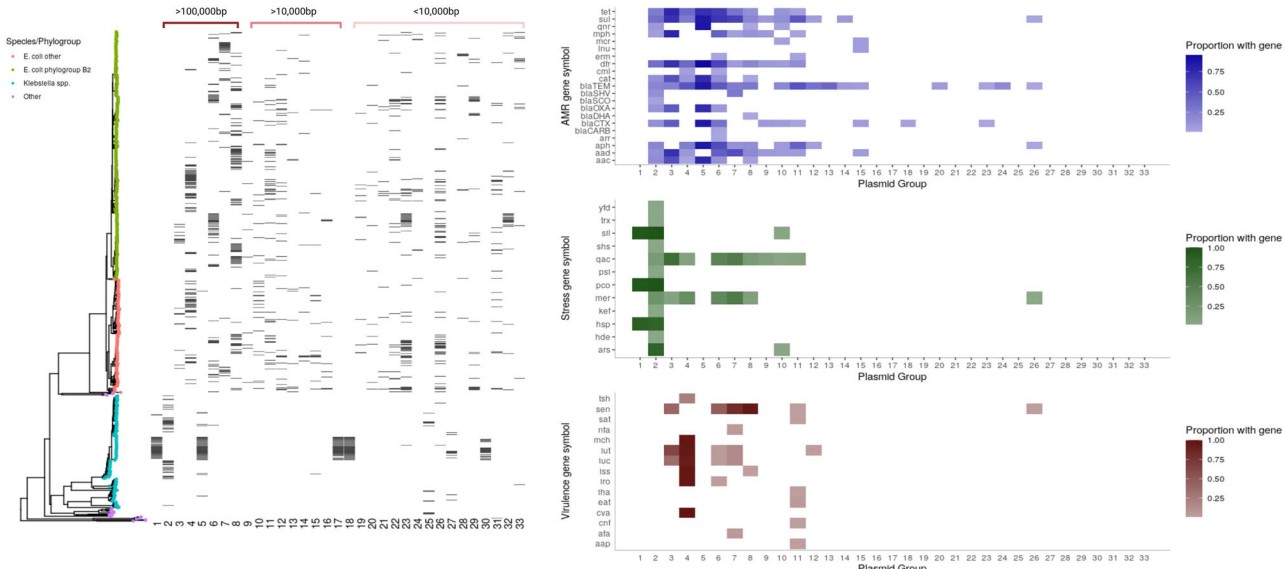

**Fig. 2 | Phylogenetic distribution of the most common ($n > = 10$ members) plasmid groups ($n = 33$ groups) and the content of these.** The tree is a neighbour-joining tree built on Mash distances between chromosomes. Tip colours represent species/phylogroup. The black bars represent the presence or absence of plasmid groups (shown along the bottom $x$ axis) for each isolate in the tree. The right panel shows the percentage of isolates within each of these 33 plasmid groups carrying the genes indicated (darker colours denote higher proportion of isolates carrying gene). To improve readability, gene groups have been clustered together. Source data are provided in the supplementary "Source Data" file.

plasmid groups (defined using a graph-based clustering method, see below) remain unsampled and that there is a significantly greater diversity amongst groups containing smaller (<100,000 bp) vs larger (≥100,000 bp) plasmids (Fig. 1B). There was some evidence that *Klebsiella* spp. isolates tended to carry slightly more plasmids than *E. coli*: median 2 (IQR 1–5) vs (median 2 (1–3) plasmids, respectively (Kruskal–Wallis, *P* value = 0.03; Fig. 1A), as did multi-drug-resistant (MDR i.e., carriage of ≥3 ARG classes) vs non-MDR isolates: ($n = 317/738$ vs. $n = 421/738$ isolates; median 3 (IQR 2–4) vs. median 2 (1–3) Kruskall–Wallis, *P* value <0.001]).

Despite comprising a relatively small proportion of the total genome (median = 2.79%, IQR 1.97–3.97%), plasmids carried 39% (2069/5311) ARGs, 12% (987/8315) virulence genes and 60% (2836/4735) stress response genes. 50% (368/738) isolates carried at least one plasmid-borne ARG and 306 at least 2; of these, 79% (242/306) carried all annotated ARGs on a single plasmid (Fig. 1C). In isolates with a medium or large plasmid, co-carriage of a small plasmid was significantly more common in isolates harbouring plasmid-borne ARGs 58% (210/361) vs. 46% (119/260) without (Fisher test, *P* value = 0.003).

### Most BSI isolates carry a large (>100,000 bp) plasmid from a small number of common plasmid groups

We first attempted to classify plasmids using existing tools; 17% (317/1880) plasmids could not be assigned a replicon type, and 33% (622/1880) had no identifiable relaxase type. Similarly, 26% (487/1880) plasmids were not typable using the recently described Plasmid Taxonomic Unit (PTU) scheme[13]; 7% (128/1880) were not typable by any method tested. Subsequently, we therefore opted to use a previously described classification approach, utilising a graph-based Louvain community detection algorithm[14] (see "Methods"), which has the advantage of not being reliant on reference databases for group assignment and is thus able to classify all plasmids into groups (hereafter referred to as "plasmid groups"). These Louvain-based plasmid groups generally clustered plasmids together at a lower distance threshold (i.e., more similar) than the other methods tests (median 0.251, IQR 0.051–0.522 vs 0.692 IQR 0.561–0.852 for COPLA/PTU clusters, 0.968 IQR 0.856–1.000 MOB-suite/Relaxase, 0.664 IQR

(0.367–0.928) Plasmidfinder/Replicon typing (Supplementary Fig. S2). This approach yielded 513 groups from 1880 plasmids, of which 164 (32%) contained >1 plasmid, but only 33 (6%) contained ≥10 plasmids, and most were singletons (349/513 (68%)). As expected, given the more closely related groupings identified by the Louvain-based approach, this method created more groups compared to the others tested, and more of these were singletons (Supplementary Data 4). In all, 322/553 (58%) *E. coli* isolates carried a plasmid from one of the four most common, predominantly *E. coli*-associated, large (>=100,000 bp) plasmid groups (4/6/7/8, all PTU-FE) in Fig. 2) and similarly 76/161 (47%) *Klebsiella* spp. isolates contained a plasmid from one of the three most common, predominantly *Klebsiella* spp.-associated, large plasmid groups (1, 2 and 5, PTU-E35, FK and FK, respectively) in Fig. 2).

### Plasmid groups are structured by host phylogeny but there is evidence of intra and inter-species transfer events

Overall, 141/513 (27%) groups were found in >1 MLST and 22/513 (4%) were found in more than one species; multi-species groups had ≥10 members significantly more commonly (8/22 (36%) vs 25/491 (5%), *P* < 0.001) (Fig. 2). We found strong evidence that the pangenome of the plasmidome of BSI isolates was structured by host phylogeny, although there was also vast and persistent background diversity. Sequence type and host species explained 8% and 7% (Adonis *P* = 0.001 for both) of the observed variance in gene content between plasmidomes, respectively. ARG content explained a comparatively small amount of variance ($R^2 = 2\%$, $P = 0.001$), as did year of isolation (0.03%, $P = 0.005$) and source attribution ($R^2 = 1.2\%$, $P = 0.99$, i.e., the suspected focus of infection, only available for a small subset of isolates [198/738]) (Fig. 3, panels a, b, c and d, respectively). When we focussed on plasmid groups found in the most common *E. coli* STs (131, 95, 73), we observed that most were seen in only a single ST (78/109), but 13 "generalist" groups were seen in all three STs, and accounted for the majority of plasmids (215/400 54%). Highly similar plasmidomes were seen in genetically divergent members of each ST, consistent with multiple horizontal transfer events (Supplementary Fig. S3). Persistent plasmid groups seen in both 2009 and 2018 were also seen in more phylogenetically diverse isolates within STs (Supplementary Fig. S4),

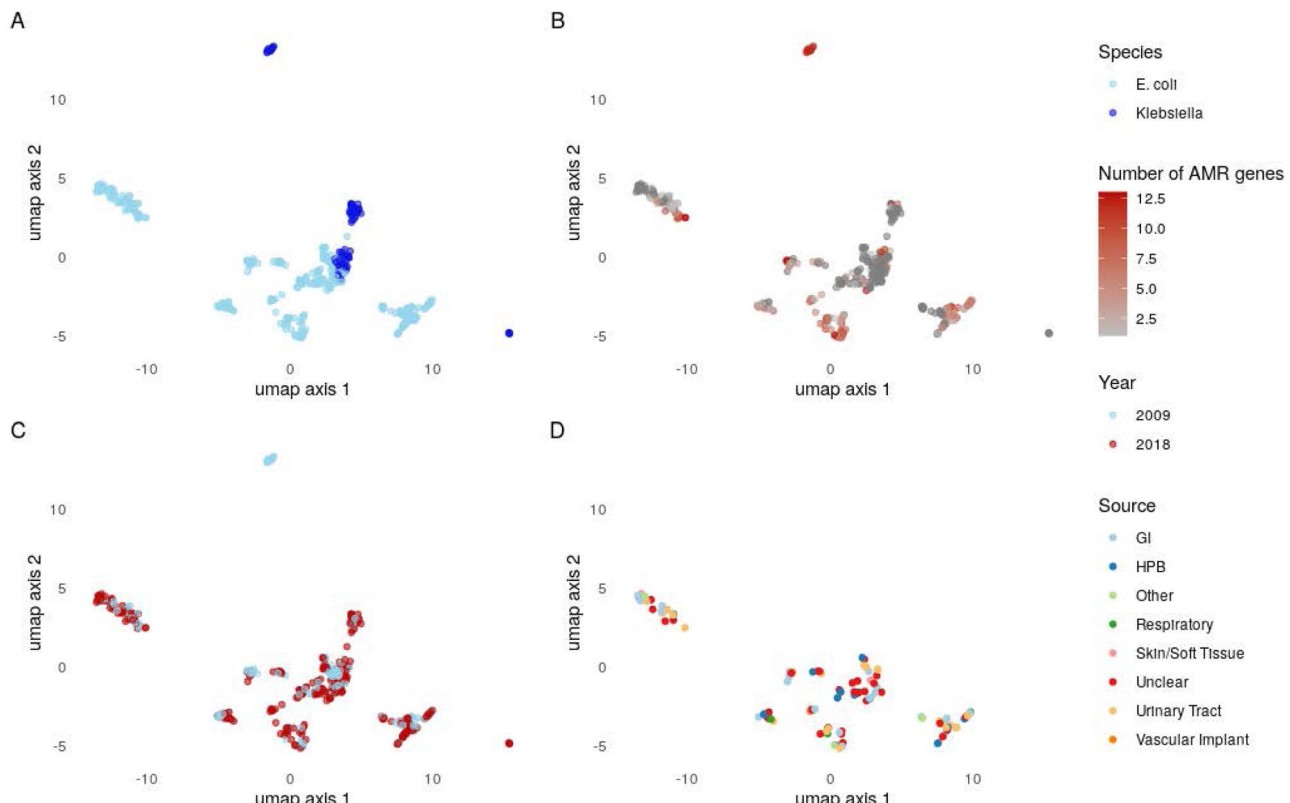

**Fig. 3 | A Umap projection of distances (measured by gene presence/absence) between the plasmidomes of isolates (each point represents the plasmidome, i.e., all plasmid sequences of a single isolate).** These are coloured to show the variability explained by species (**A**)/ARG carriage (**B**)/year (**C**) and infection source (**D**). Source data are provided in the supplementary "Source Data" file.

suggesting that the persistence of plasmids may be linked to their host range potential.

### Common plasmid groups share genes with each other; gene sharing with chromosomes is also frequent

Whilst we observed only 4% (22/513) plasmid groups were shared between species, we hypothesised that this might greatly under-represent the true extent of plasmid-mediated horizontal gene transfer given the role of smaller mobile genetic elements and the fact that BSIs represent a tiny fraction of the overall ecological landscape. We therefore looked for evidence of overlap in the pangenome between different plasmid groups as well as between these and host chromosomes. Most genes in the pangenomes of common (i.e., containing $n \geq 10$ plasmids) plasmid groups of *E. coli* and *Klebsiella* spp. were non-unique to their group (median % non-unique genes 88%, IQR 67-98%). Most overlap occurred amongst genes found in the plasmid pangenome from the same species (median % shared genes 86% (IQR 50–95%) vs 31% (8–43%) from different species, $P < 0.001$). There was also substantial overlap between plasmid group pangenomes and the chromosome pangenome, although there was some evidence of convergence in the chromosomally integrated mobilome between species, evidenced by less difference in the proportion of genes shared with the chromosome for the same vs different species (Supplementary Fig. S5, median 33% (IQR 0–45%) vs 21% (0–35%), respectively $P = 0.34$).

### Plasmids associated with ARG carriage are often highly similar to those with no such genes

The 439 plasmids carrying at least one ARG were predominantly large ($\geq 100,000$ bp, 277/439, 63%), low copy number (median 1.80 IQR 1.63–2.37) and conjugative (347/439, 79%). Whilst most plasmid-borne ARGs were carried by plasmids clustering in a small number of groups (i.e., 81% 1674/2069 ARGs were carried by 8 plasmid groups), 36% (170/ 474) plasmids in these groups did not carry an ARG and all groups had at least one such member, highlighting that acquisition of ARGs in ARG-negative plasmid backbones represents a common risk across genetically divergent plasmid groups (Fig. 4). We repeated this analysis using group assignments given by COPLA (Plasmid Taxonomic Units) and Plasmidfinder (replicon typing) and found similar results (Supplementary Data 5), suggesting that this finding is robust to the choice of clustering method.

### Hybrid assembly reveals complex nested diversity associated with key AMR genes, significant chromosomal integration of ARGs and the presence of multiple copies in different contexts

Chromosomal integration of ARGs was common: for example, in *E. coli*, 56% (23/41) $bla_{CTX-M-15}$, 9% (2/22) $bla_{CTX-M-27}$, 14% (42/293) $bla_{TEM-1}$, 42% (14/33) $bla_{OXA-1}$, 39% (7/18) $aac(3)$-$IIa$ and 5% (3/65) $dfrA17$ were chromosomally integrated. There was significantly more chromosomal integration of ARGs also seen at least once in a plasmid in our study in *E. coli* vs *Klebsiella* spp. (restricting to 2009 and 2018 only 15% [324/2103] vs 8% 39/478 [8%], Chi-squared test $P < 0.001$). For *E. coli*, there was significantly more chromosomal integration in 2018 vs 2009 (19% 285/1485 vs 6% 39/618, Chi-squared test $P < 0.001$) but there was no evidence of this for *Klebsiella* spp. (7% [3/190 vs 6% 17/279, Chi-squared test $P = 0.89$). For most of these ARGs, there were multiple instances of isolates carrying two (and occasionally more) copies (9 such examples for $bla_{CTX-M-15}$ (Fig. 5), 1 $bla_{CTX-M-27}$, 29 $bla_{TEM-1}$, 2 $aac(3)$-$IIe$ and 1 $dfrA7$).

Given the global importance of the ESBL gene $bla_{CTX-M-15}$ conferring third generation cephalosporin resistance, we focused on its genetic background and putative dissemination mechanisms. As mentioned above, plasmid groups carrying this gene in our dataset were generally species-constrained. However, within a single species, considering phylogroup, sequence type and even plasmid group,

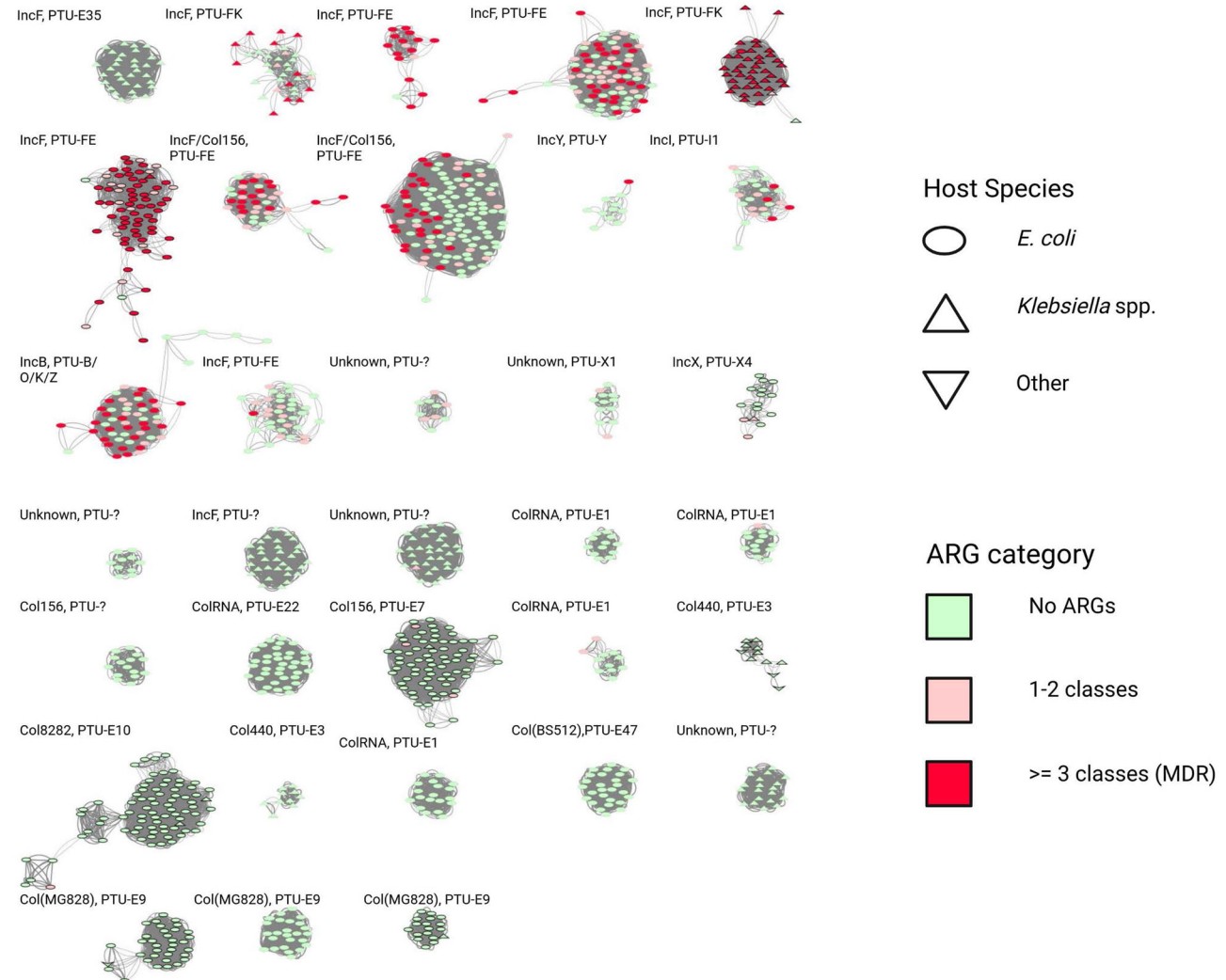

**Fig. 4 | Plasmid network where individual plasmids (nodes) are connected by edges if they cluster in the same group using the Louvain-based methodology and coloured according to the number of classes of ARGs that they carry.** Edge thickness is drawn proportional to the Jaccard distance (see methods) between plasmids. Multi-species clusters are donated by black outlined shapes. Only plasmids groups with ≥10 members are shown. The ordering of clusters corresponds to that in Fig. 2. Labels above clusters denote the PlasmidFinder/COPLA taxnomic designations, respectively; plasmid groups are numbered consecutively from the top left. Source data are provided in the supplementary "Source Data" file.

$bla_{CTX-M-15}$ was found in a variety of genetic contexts (Fig. 5). For example, in *E. coli* ST131 it was found in five plasmid groups and was chromosomally intergrated in 41% (17/41) isolates. Within ST131 subclades, there was some evidence of vertical transmission, as well as numerous independent integration events. In many cases, several unique gene flanking regions were found in association with $bla_{CTX-M-15}$ within a single plasmid group, or identical flanking regions were shared across plasmid groups and between plasmid groups and chromosomes. Visual inspection of gene flanking regions and hierarchical clustering of a weighted graph ("Methods": Bioinformatics) suggested that whilst there was substantial diversity, these flanking regions appear to have evolved in a stepwise manner with bilateral association of $bla_{CTX-M-15}$ and Tn2 in flanking groups 2, 3 and 6 compared to the presence of Kpn14 (groups 1 and 5) and IS26 (group 4) (Supplementary Fig. S6). Inspection of core-genome phylogenies of the two largest $bla_{CTX-M-15}$ carrying plasmid groups (plasmid groups 2 [IncF, PTU-FK] and 3 [IncF, PTU-FE] in Fig. 2) demonstrated multiple probable independent horizontal acquisition events of transposable units containing this gene (and other ARG cassettes Supplementary Figs. S7 and S8), suggesting that a flexible capacity to acquire ARGs through diverse mobile genetics elements rather than a fixed association with them might be important factors for the successful dissemination of the host plasmid.

## Comparison with wider plasmid datasets highlights undersampled plasmid diversity, more widespread inter-species and inter-niche plasmid sharing, and the potential for carbapenemase dissemination amongst high-risk plasmid groups

We repeated our graph-based plasmid clustering method on a combined dataset of Oxfordshire plasmids (N = 1880, hereby referred to as the "Oxfordshire dataset") and the Global collection of plasmids deposited in the NCBI (N = 10,159, denoted the "global dataset") using the same sparsifying threshold (≤0.551). This yielded 5913 groups, of which 484 contained at least one plasmid from the "Oxfordshire dataset"; of these, 326/484 groups (67%) containing 536 plasmids appeared to be unique to Oxfordshire. In total, 79/484 (16%) of groups containing Oxfordshire plasmids were found in more than one species in the full dataset; of these 57 (72%) occurred in only a single species in the Oxfordshire dataset, highlighting the substantial underestimation of wider between-species dissemination by investigating only a single region and single source (i.e., bloodstream infections).

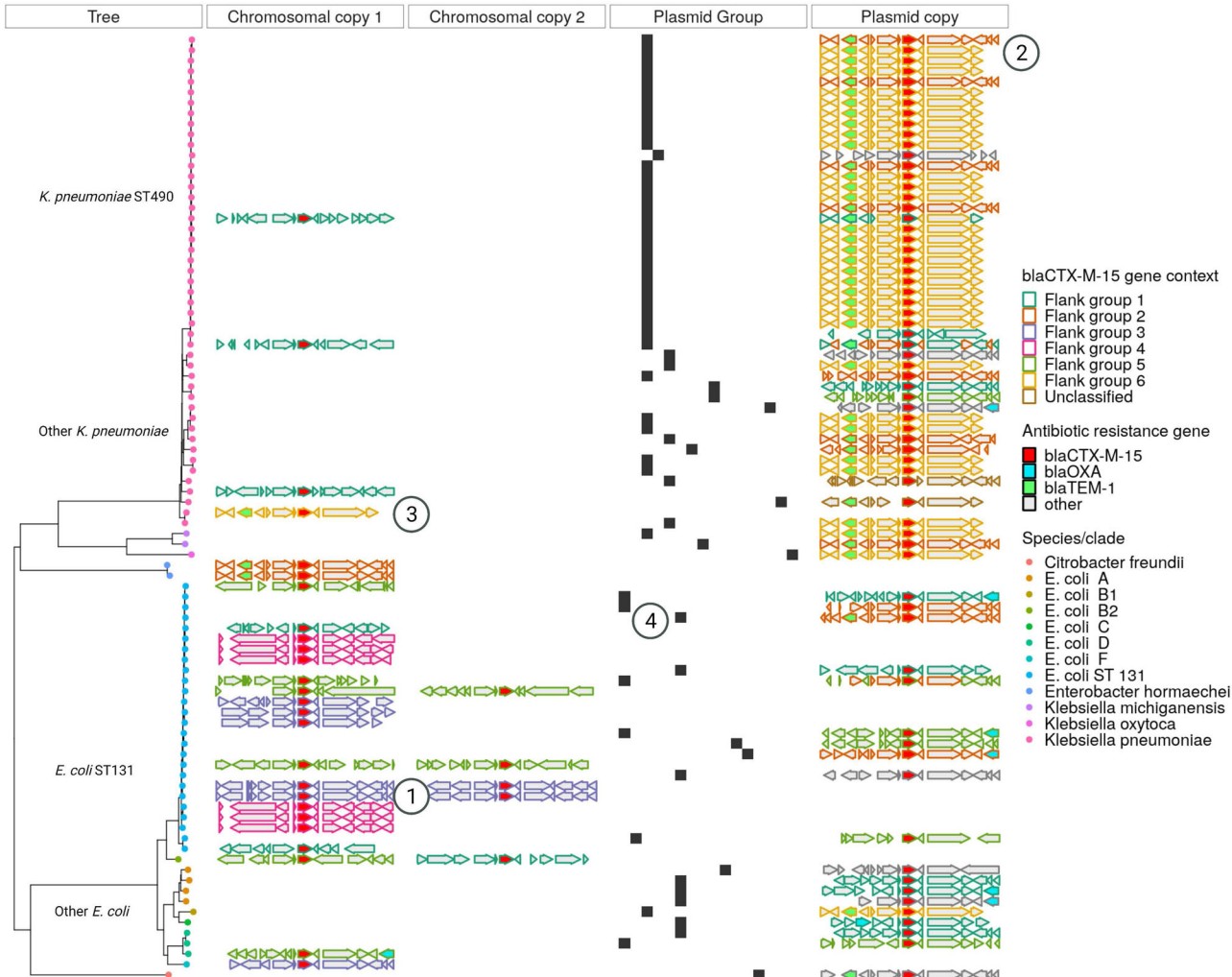

**Fig. 5 | Nested genetic complexity associated with *bla*$_{CTX-M-15}$ mobilisation.** The "Tree" panel shows a neighbour-joining tree of Mash distances between chromosomes for isolates carrying a *bla*$_{CTX-M-15}$ gene. Tip colours represent species/ST/phylogroup. The chromosomal copy 1 and 2 panels show the genetic context 5000 bp up- and downstream from chromosomal copies of the *bla*$_{CTX-M-15}$ gene (shown in red); the plasmid copy panel shows this equivalent information for isolates carrying a plasmid-borne copy of this gene. The outlining colour in these panels shows the hierarchical cluster assignment of these flanking groups. The plasmid group panel shows group membership of plasmids carrying the *bla*$_{CTX-M-15}$ gene with each *x* axis position representing a distinct group and black bars showing the presence or absence of these for isolates in the tree. The encircled numbers denote: 1−different flanking sequences in the same ST, 2−different flanking sequences in the same plasmid group, 3−the same flanking group found in both chromosomal and plasmid contexts and 4−different plasmid groups harbouring the gene found within the same ST. Source data are provided in the supplementary "Source Data" file.

A striking feature of the global network was that plasmids carrying carbapenemase genes clustered with those that did not (Fig. 6). Of 122 plasmid groups with at least one member carrying a carbapenemase gene, 19 (16%) contained at least one Oxfordshire plasmid. These included representatives from the *K. pneumoniae* MDR-associated Oxfordshire BSI dataset groups 2 and 5 (Fig. 2), three large groups (Fig. 2, groups 3/6/8) widely distributed amongst E. coli isolates and two groups of smaller plasmids (<100,000 bp, Fig. 2 groups 10 and 12), also widely distributed in Oxfordshire *E. coli*. Although only 2% (7/414) Oxfordshire plasmids falling into these groups actually carried a carbapenemase ARG, this suggests the potential for carbapenemase acquisition and dissemination amongst widespread "high-risk" plasmid backbones.

### Factors predictive of plasmid group success
Having demonstrated that most isolates carry a plasmid from a relatively small number of plasmid groups, we next sought to understand what factors might be driving the widespread dissemination of these amongst BSI isolates. Multivariable Poisson regression analysis revealed that plasmid group frequency (a subjective marker of

evolutionary "success") was associated with isolation in multiple species (adjusted rate ratio aRR 4.89, 95% CI 4.29−5.57, *P* < 0.001), capacity to conjugate (aRR 1.73, 95% CI 1.47−2.04) or mobilise (aRR 1.29, 95%CI 1.13−1.48) (i.e., containing either a relaxase or *oriT* but missing a mate-pair formation marker), carriage of multiple ARGs (aRR 1.23, 95% CI 1.19−1.27)/virulence (aRR 1.44, 95%CI 1.36−1.53), toxin−antitoxin genes (aRR 1.32, 95% CI 1.18−1.47) and a higher GC content (aRR 1.01, 95% CI 1.00−1.03) (Supplementary Data 6). Carriage of ARGs (adjusted odds ratio, (aOR = 2.88, 95% CI 1.53−5.41, *P* < 0.001) and isolation in multiple species (aOR = 7.79, 95% CI 3.07−22.90, *P* < 0.001) were independently associated with a higher probability of plasmid groups being observed internationally (Supplementary Data 7).

### Machine learning allows risk stratification of plasmids
Given that we have shown that plasmids carrying ARGs are often very similar to those with no such genes ("ARG-negative plasmids"), we hypothesised that it might be possible to predict whether ARG-negative plasmids pose a risk for eventual association with ARGs. To do this, we first performed a genome-wide association study using the

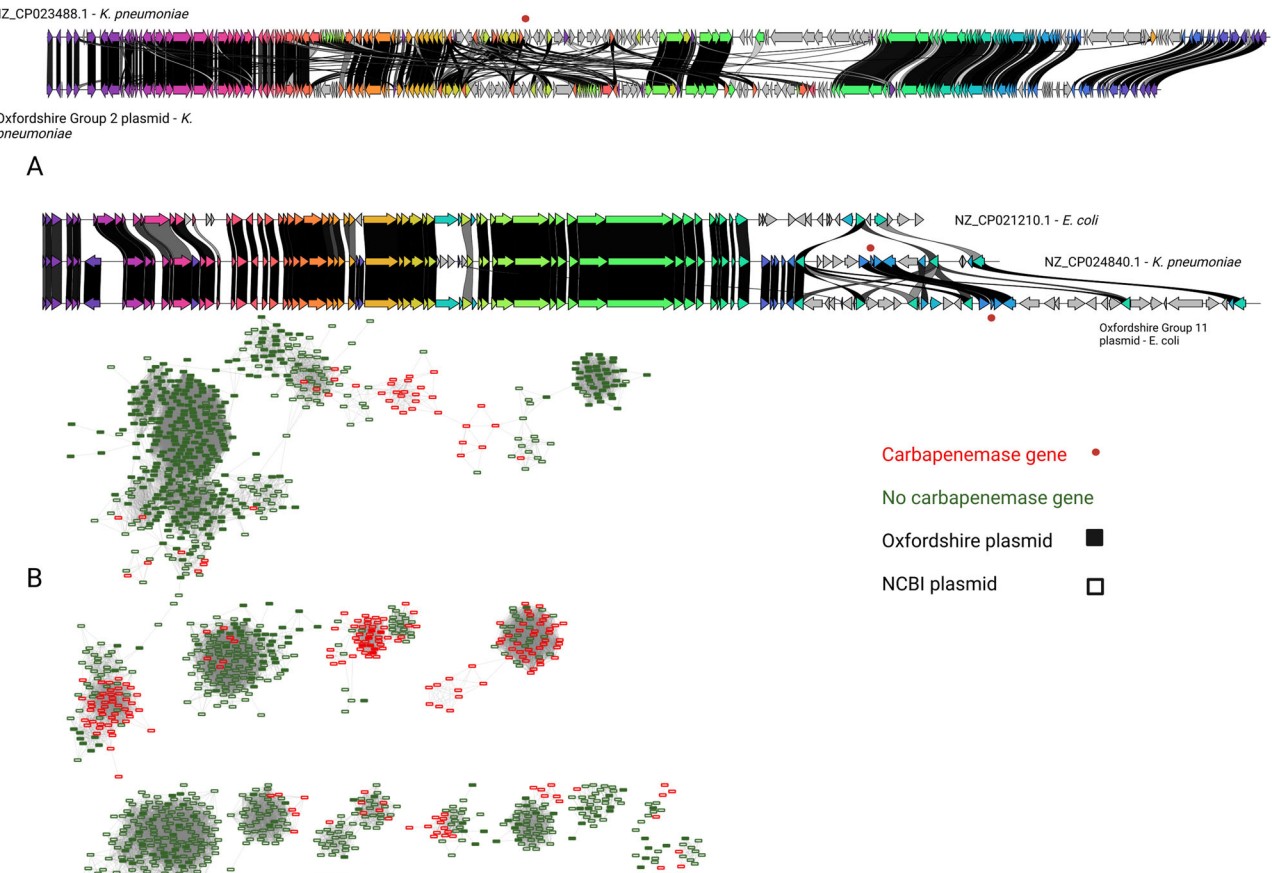

**Fig. 6 | Plasmids carrying carbapenemase genes are highly similar to plasmids without these genes found in Oxfordshire BSIs. A** Each horizontal bar represents a plasmid assembly either from Oxfordshire "Group 2/11" or the NCBI global dataset. Common genes are shown in colour (with blast identity between these shown from light grey to black (where the latter represents a perfect match), whereas genes unique to a given plasmid are shown in grey. **B** A network plot of plasmids which cluster with carbapenemase-carrying plasmids in the global network analysis. The "carbapenemase gene" grouping includes all those variants identified and classified as conferring resistance to the class "Carbapenem" in AMRFinder. Plasmids (nodes) are connected with an edge where the edge weight is ≤0.551 (see "Methods"). The thickness of the edges is displayed so that it is proportional to the edge weight. Source data are provided in the supplementary "Source Data" file.

Oxfordshire dataset only, which was corrected for population structure and plasmid size, to identify genes (excluding known ARGs) significantly more or less likely to be carried by plasmids in ARG-associated groups (i.e., plasmid groups where at least one member carries at least one ARG). This revealed significant associations between ARG-associated plasmid groups and the presence of insertion and transposon sequences, various virulence factors, toxin/antitoxin system and heavy metal resistance genes (Supplementary Data 8).

We then tested the predictive value of these elements to identify ARG-negative plasmids belonging to ARG-associated groups using a variety of models on the Oxfordshire dataset (see "Methods") with stratified tenfold cross-validation to estimate out-of-sample performance. The best-performing model (Random Forrest) had a mean accuracy of 90.3% (standard deviation [SD] 2.4%), mean area under the receiver operator curve [AUC] 0.90 (SD 0.02), mean sensitivity 86% (SD 4.3%) and mean specificity 93.4% (SD 2.9%). We re-trained the Random Forrest model on ARG-negative plasmids in the global dataset using only plasmids sequenced prior to 2018 and subsequently made predictions on the held-out 2018 plasmids. This demonstrated that the model generalised well but was less sensitive on this dataset (accuracy 84.6%, AUC 0.82, sensitivity 73.7% and specificity 89.9%).

## Discussion

In this study, we fully reconstructed 738 isolates (1880 plasmids) to conduct a comprehensive evaluation of the epidemiology and function of plasmids associated with Gram-negative isolates causing bloodstream infections to date. Most isolates in this study carried a large plasmid from a small number of plasmid groups; these were frequently, but not invariably, associated with the carriage of multiple antibiotic resistance, virulence and heavy metal resistance genes, potentially providing survival and fitness benefits to the host bacterium. The fact that most isolates with multiple plasmid-borne ARGs (often from several different classes) carry all of these on a single plasmid reinforces the importance of good antimicrobial stewardship and avoiding unnecessary exposure to all classes of antibiotics to control co-selection as much as possible. Crucially we also found that plasmids carrying ARGs frequently cluster in large, widely disseminated groups with plasmids without these genes, representing a potential set of "high-risk" backbones for ARG acquisition and horizontal spread.

To date, most similar sequencing studies have focused on plasmids carrying particular ARGs (particularly those with ESBL/carbapenem resistance genes) and have therefore not considered how these might be related to plasmids without such genes. We hypothesise that plasmid adaptation to co-exist with successful lineages often occurs prior to the acquisition of high-risk ARGs, presenting a potential window of opportunity for intervention which is lost if one is solely focused on the presence of these genes. Our data and should therefore motivate a shift away from studies focusing on a single phenotype or gene of interest and towards efforts to identify and track high-risk

plasmid groups or other smaller mobile genetic elements and clearly illustrate that such a new surveillance framework must incorporate unselected sampling frames, i.e., not only selecting isolates with particular AMR phenotypes for sequencing. Notably we found an association of small plasmids and medium/large ARG-associated plasmids, suggesting that they may play an important helper role in ARG plasmid persistence/spread, and a more detailed understanding of this possible synergy could be valuable[15].

Whilst plasmid populations were structured, and plasmid groups were mostly constrained to a single species and in some cases species lineages in Oxfordshire, there was also clear evidence of exchange between lineages of a species and different species. The plasmid groups we identify appear to be subsets of the recently described PTU typing system (e.g., plasmid groups 3/4/6/7/8 in Fig. 4 all correspond to PTU-FE) and there appears to be a non-random distribution of these across the phylogeny in Fig. 2, possibly indicating that these subsets are adapted to particular sequence types (or local/ecological niches). Enterobacterales are widely distributed as commensals and in multiple environmental sources; our study sample is thus extremely sparse relative to the whole ecology. Even where we found no evidence that certain plasmids were shared between species in Oxfordshire, our data demonstrated widespread sharing of the plasmid gene repertoire (including ARGs and their flanking regions) with plasmids and chromosomes in other species. The high proportion of isolates with chromosomally integrated ARGs (and apparent increase across the study period for E. coli) supports findings from previous studies[16,17] and may either represent a success of plasmids in conferring survival benefits to their host while lowering their own associated fitness cost or a success of the host by lowering its dependence on the presence of the plasmid. It may also reflect their frequent association with transposable units[6] (e.g., *ISCEp1/IS26* for $bla_{CTX-M-15}$) which facilitate their integration and dissemination.

A limitation of this study is that it is from a single region, mitigated in part by comparisons with publicly available datasets. We chose to exclude assemblies with non-circularised plasmids, which may exclude linear plasmids, though these are difficult to differentiate from incomplete circular plasmids. The species imbalance in the dataset reflects the fact that E. coli BSIs are more common than those caused by *Klebsiella* spp. but may nevertheless bias inferences about plasmid and gene sharing within and between species. Whilst the decision to include an enriched sample (representing 26% of the total dataset in this study) can bias the data with over-representation of dominant clones, it is also true that such clones dominate the population structure of Oxfordshire BSI isolates (and global isolates in the case of E. coli[12,18,19]). The lower sensitivity for predicting ARG-group association in the global dataset likely reflects its inherent bias and heterogeneity compared to the Oxfordshire dataset as well as the existence of such ARG-associated groups and genes not observed in our setting. The inability to sequence and/or assemble all plasmids from the selected cohort is an additional limitation. Sequencing only bloodstream infection isolates may lead to underestimation of how much sharing of plasmids between species truly occurs given that this represents a highly selected subset of isolates causing severe disease. This is supported by our analysis of the large publicly available dataset, which demonstrated that several groups found only in a single species in our study have previously been seen in other species. Our results also highlight the substantial limitations of previous studies using reference database-based approaches for plasmid typing and demonstrate that fully reconstructed genomes (i.e., long-read sequencing data) are essential in order to provide meaningful insight.

In conclusion, our study provides a high-resolution description of the plasmidome associated with E. coli/*Klebsiella* spp. bloodstream infections and demonstrates that using long-read data and unselected sampling frames is essential in order to fully appreciate its complexity. Previous studies of plasmid epidemiology in Gram-negatives have

primarily focused on MDR/carbapenemase-carrying isolates; our finding that non-ARG-carrying plasmids are often highly similar to plasmids isolated in these earlier studies demonstrates the potential for rapid dissemination of ARGs to settings where they are currently rare. We recommend that surveillance is based on unselected sampling frames, long-read sequencing and considers plasmids and smaller mobile genetic elements to develop a representative understanding of the horizontal gene transfer landscape to facilitate appropriate intervention.

## Methods

### Isolate selection
The Oxford University Hospital Foundation Trust microbiology laboratory provides a service for four general/specialist referral hospitals as well as all GP practices (community/primary care) in the region (total catchment population 805,000). We have previously reported analyses of short-read sequencing data from E. coli and *Klebsiella* spp. bloodstream infection isolates in Oxfordshire between 2009 and 2018[12,20]. In this earlier study, we sequenced all available isolates in this time period (i.e., a non-biased, sequential and near complete dataset; $n = 3468$ isolates) with de-duplication to 90 days per patient. In the current study, we additionally sequenced all E. coli and *Klebsiella* spp. isolates from 2009 and 2018 using Oxford Nanopore Technologies (547/738, 74% isolates successfully sequenced, Supplementary Data 1, Supplementary Fig. S1). We also sequenced a subset of isolates from intervening years, using stratified random sampling based on analysis of short-read data to capture maximum plasmid diversity. To make this selection, we analysed the contigs of all remaining short-read assemblies using MLPlasmids[21] to classify then as likely plasmid or chromosomal in origin (using -use-full-khash-sets). All likely plasmid contigs where binned together and sketched using Dashing[22] (default settings), and a distance matrix was subsequently collected. We sparsified this matrix at 0.8 and used the LinkComm[23] package in R to identify communities. We then selected one representative per ST from the largest ($n \geq 10$) clusters. The remaining capacity (limited by resource and time as laboratory work took place during the SARS-COV-2 pandemic) was filled using isolates with similar multi-species plasmidomes and local AMR-associated outbreak clones. Of the 191/738 (26%) isolates selected as part successfully sequenced as part of this enriched dataset, 17/191 (9%) were *K. pneumoniae* ST490 (the dominant ST in Oxfordshire over this time period[12]) and 55/191 (29%) belonged to the predominant E. coli sequence types (STs 131/95/73/69). A breakdown of successfully sequenced isolates and those excluded is shown in Supplementary Fig. S1.

### Sequencing
DNA for long-read sequencing was extracted either using Qiagen Genomic Tip/100G according to the manufacturer's instructions, or with the BioMerieux Easymag using the manufacturer's generic short protocol with a final elution volume of 50 µL. The Qubit 2.0 Flourometer was used to quantify DNA. Sequencing libraries were prepared using the Oxford Nanopore Technologies Native ($n = 23$) and Rapid (all other) barcoding kits (SQK-RBK004, SQK-LSK108), according to the manufacturer's instructions. Sequencing was performed on GridIons with R9.4 flowells, which were reused multiple times utilising the ONT Flow Cell Wash kit and our previously validated protocol[24]. For short-read sequencing, DNA was extracted using the QuickGene DNA extraction kit (Autogen, MA, USA) as per the manufacturer's instructions with the addition of a mechanical lysis step (FastPrep, MP Biomedicals, CA, USA; 6 m/s for 40 s). Short-read (150 bp) sequencing was performed as part of a previous project[20] on HiSeq (Illumina instruments.

### Bioinformatics
All bioinformatic programmes were run using default settings unless otherwise specified. Reads were first base-called and demultiplexed

using Guppy (v3.1.5, Oxford Nanopore Technologies) with Deepbinner[25] (v0.2.0) subsequently used to recover additional unclassified reads[24]. Our strategy for hybrid assembly is depicted in Supplementary Fig. S9. We first assembled all isolates using Unicycler[25] (v0.4.9) (--mode bold) with the raw Illumina and ONT reads as input. In parallel, we performed another assembly where Unicycler was given an assembly graph from Flye[26] (v2.9-b1768 run with --plasmids --meta and reads which had been polished using Ratatosk[27] v0.7.6.3) and short reads pre-processed by Shovill[28] (v1.1.0). The most contiguous assembly of these was used (or the latter if both were complete). If neither hybrid assembly completed then we used the Flye assembly (with four subsequent rounds of Pilon[29] (v1.24) polishing) if this was complete. Incomplete assemblies (where ≥1 replicon [i.e., either plasmids or the chromosome] had >1 contig) were excluded from further analysis ($n = 215$). Basic assembly metrics can be found in Supplementary Data 1.

Rarefaction analysis with performed using the R library Micropan[30]. Annotation of genes was performed using AMRFinder Plus[31] (v3.10.23) (Supplementary Dataset 2), ABRicate[32] (v1.0.1), TADB 2.0[33] and Prokka[34] (v1.14.6; custom/manually augmented databases (available at www.github.com/samlipworth/GN_BSI_Hybrid) were used for the latter two to attempt to improve the proportion of annotatable toxin–antitoxin systems/plasmid-associated genes respectively. GC content and predicted mobility were extracted from MOB-suite (v3.0.0) output. GC-gap was defined as *GC content of plasmid - GC content of chromosome*. Unicycler was used to estimate plasmid copy number.

Plasmidome pangenomes (where all plasmids carried by an isolate are considered as a single unit) were analysed using Panaroo[35] (v1.2.8 --clean-mode sensitive --family_threshold 0.7) and visualised with a Umap projection created using the R package Umap[36]. Variance in the pangenome explained by e.g., AMR content/year/species was examined using a permanova perfomed in the R package vegan[37]. Gene flanking regions were analysed using Flanker (v1.0, -w 0 -wstop 5000 -wstep 100)[38]. The Reder package[39] was used to cluster a weighted graph created from a matrix in which distances were determined to be the greatest distance from the gene (in both upstream and downstream directions) in pairs of isolates which were in the same Flanker cluster; this analysis was repeated in an all vs all fashion for all isolates. Flanking regions were annotated using the Galileo AMR software (Arc Bio, Cambridge, MA, USA).

## Plasmid clustering

Robust taxonomic classification of plasmids remains a challenge[40]. We therefore used two established methods that have been applied to large-scale short-read sequencing datasets, Replicon typing using PlasmidFinder (--minid 80)[41] and Relaxase typing with MOB-suite (default settings i.e., --min_mob_cov 80)[42]. We also typed all plasmids using the recently described Plasmid Taxonomic Unit nomeclature[11] (using COPLA[13]). As a substantial number of plasmids remained unclassified by all these methods, we additionally utilised a recently described graph-based classification system[14]. Mash(v2.3)[43] (-s 1000, -k 21) was used to create an all vs. all distance matrix of plasmid assemblies where the distance was taken to be 1−the proportion of shared kmers between the plasmid of interest and plasmids in the sketch sequences, where plasmids with a distance of 0 share all kmers in the sketch space whereas those with a score of 1 share no common kmers. This was used to create a weighted graph using the R package Igraph[44] where vertices represent plasmids and edges between these are weighted by the distance described above. Community detection on this graph was performed using the Louvain algorithm which seeks to maximise the density of edges within vs between communities. We optimised the performance of this algorithm as described previously[14] by sparsifying the graph, removing edges with a weight ≤ a threshold which was selected by iteration. This approach performed optimally

(i.e., assigned the maximum number of isolates to larger [$n ≥ 10$ isolate] clusters) when the graph was sparsified at an edge weight of ≤0.551 prior to community detection (Supplementary Fig. S10); this parameter was used for all subsequent analysis. The final sparsification threshold was selected to optimise the number of plasmids assigned to large ($n ≥ 10$) clusters. We compared the classifications given by this approach to other methods using the Normalised Mutual Information index in the R package NMI[45], which demonstrated good agreement with previously described classification methods using normalised mutual information (NMI; see "Methods"): replicon-typing NMI=0.81, relaxase-typing NMI=0.93, plasmid taxonomic unit (PTU) NMI=0.81.

## Comparison with existing plasmid sequencing data

To place our plasmid sequencing data in a global context, we downloaded all available plasmids ($n = 10,159$) from a recently curated plasmid collection[10] for comparison. We refer to the Oxfordshire isolates as the "Oxfordshire dataset" and the combined collection as the "Global dataset". We computed a pairwise distance matrix and performed Louvain-based clustering as described above, sparsifying the graph using the same threshold (0.551) as in the main analysis.

## Statistical analysis

To investigate factors associated with the geographical dissemination of plasmid groups, we subsetted the global dataset to include only Oxfordshire isolates and those from NCBI not from the UK and where the location of isolation was known. We further filtered this to include only plasmid clusters observed at least once in Oxfordshire. Isolation in more than one country was used as the binary dependent variable in a logistic regression with other plasmid group features (e.g., ARG/virulence/GC content) as independent variables. Multivariable associations between all available plasmid group metrics (independent variables) and plasmid group frequency in the dataset (dependent variable) were estimated using Poisson regression in exploratory analyses. Comparisons of continuous variables and proportions between groups used Kruskal–Wallis/Wilcoxon rank-sum and Fisher/Chi-squared tests respectively in R version 4.1[46].

To search for non-AMR plasmid-borne genes associated with carriage of ARGs, we performed logistic regression with membership of an ARG-associated group as the dependent variable and each gene in the plasmid pangenome as the independent variable, adjusting for population structure using multi-dimensional scaling (MDS) of mash distances (R package CMD scale), represented in ten dimensions[47]. We additionally adjusted for plasmid size using three categories ("large" ≥100,000 bp, medium ≥10,000– < 100,000 bp and small <10,000 bp). P-values were adjusted for multiple comparisons using the Bonferroni method after removing genes with <1% population frequency. This feature selection performed using a pangenome-wide association study was conducted using only the Oxfordshire plasmid dataset.

We then tested the predictive value of these genes ($n = 178$) to identify plasmids not carrying ARGs (ARG-negative plasmids) which were found in ARG-associated groups in the Oxfordshire dataset ($N = 1439$ ARG-negative plasmids of which 609 were in ARG-associated groups). We evaluated the performance of nine models (logistic regression, linear discriminant analysis, K neighbours classifier, decision tree classifier, gaussian naive Bayes, random forest classifier and gradient boosting classifier and a voting classifier combining all of these) using 10-fold cross-validation, which was repeated 100 times. We further evaluated the performance of the best-performing model (random forest) using ARG-negative *E. coli* and *Klebsiella* spp plasmids in the global dataset with the same features as before (significant gene hits from the pangenome GWAS above). We split the dataset into plasmids collected prior to 2018 (global training set on which the model was re-trained $n = 656$ plasmids of which 221 where in ARG-associated groups) and those collected subsequently (held-out global testing set on which final metrics were reported $N = 306$ plasmids of

which 99 where in ARG-associated groups). This analysis was performed using the SciKitLearn[48] package in Python version 3.7.7.

## Data visualisation
Data were visualised using the ggplot2[49] and gggenes[50] packages in R, Clinker[51], Cytoscape[52] and Biorender (www.biorender.com).

## Reporting summary
Further information on research design is available in the Nature Portfolio Reporting Summary linked to this article.

## Data availability
Sequencing data for the "Oxfordshire dataset" has been deposited under NCBI project accession PRJNA604975 and at https://doi.org/10.25452/figshare.plus.24573268. Sequencing data for the global dataset (https://doi.org/10.1038/s41467-020-16282-w) is available from the NCBI RefSeq repository ftp://ftp.ncbi.nlm.gov/refseq/release/plasmid. Source data are provided with this paper.

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

## Acknowledgements
This study is supported by the National Institute for Health Research Health Protection Research Unit (NIHR HPRU) in Healthcare Associated Infections and Antimicrobial Resistance at the University of Oxford in partnership with Public Health England (PHE) (NIHR200915). W.M. is supported by a scholarship from the Medical Research Foundation National PhD Training Programme in Antimicrobial Resistance Research (MRF-145-0004-TPG-AVISO). A.S.W. and T.E.A.P. are also supported by the NIHR Oxford Biomedical Research Centre. A.S.W. is an NIHR Senior Investigator. N.S. is an NIHR Oxford BRC Senior Fellow. The views expressed are those of the authors and not necessarily those of the National Health Service, NIHR, Department of Health, or PHE. S.L. is supported by an MRC Clinical Research Training Fellowship (MR/T001151/1). L.P.S. is a Sir Henry Wellcome Postdoctoral Fellow funded by Wellcome (Grant 220422/Z/20/Z). Computation used the Oxford Biomedical Research Computing (BMRC) facility, a joint development between the Wellcome Centre for Human Genetics and the Big Data Institute supported by Health Data Research UK and the NIHR Oxford Biomedical Research Centre. The authors gratefully acknowledge the assistance of Anthony Brown in performing DNA extractions.

## Author contributions
S.L., T.P., D.C., A.S.W. and N.S. conceptualised the study and obtained funding. Laboratory work was performed by S.L., G.R., K.C., L.B., S.G., A.V. and J.K. M.A., M.M., K.J. and S.O. are responsible for supervision and oversight of the clinical laboratory. Analysis was performed by S.L., W.M. and L.S. with additional input from K.D.V. and T.D. The overall supervision for the project was provided by T.P., D.C., S.H., A.S.W. and N.S. S.L. wrote the first draft of the manuscript, and all authors contributed to subsequent revisions and reviewed the final manuscript prior to submission.

## Competing interests
The authors declare no competing interests.
