## [Peer Review File · Nature Communications]

The plasmidome associated with Gram-negative bloodstream infections: A large-scale observational study using complete plasmid assembliesReviewer #1 (Remarks to the Author):

The manuscript by Lipworth et al provides an apparent exhaustive analysis of the plasmid sequences from isolates of Enterobacterales caused by bloodstream infections (BSI) in a single hospital. The isolates were collected in a timespan of 10 years but most of them correspond to 2009 and 2018. The amount of sequencing and bioinformatic analysis performed is impressive but the novelty of the findings is very limited, and the interpretation of the results deserves a deep and careful revision. All the results that the authors show as "novel" (eg. the notion of high-risk plasmids) or as unexpected (e.g. high rate of the chromosomal location of blaCTX-M-15) are well-known by people working in the field of epidemiology of bacterial antimicrobial resistance. The lack of categorization of the plasmids according to traditionally typing systems (replication, mobilization) and the weak connection between figures and some sections of the paper make it impossible to understand what the results really mean beyond the description of different predominant plasmid groups which may or not carry ARGs (what is the novelty of this?). Despite the amazing and huge available information about some plasmid families and the transmission pathways of bla genes -which is relevant to interpreting and discussing what is explained in the paper-, most of these key publications are missing. This made some statements appear as "novel" or "surprising" when they are not. Finally, it would be appropriate what this publication adds to the previous ones by the authors (ref 11 and 12). Some results could be redundant (clonal background, plasmid diversity).

Some comments are given for the author's consideration.

Major comments.

- **The sample.** The criteria to select the isolates should be described to properly interpret the results. The authors have previously published two papers (references 11, 12 which support the 2009 and 2018 collections of isolates) but the reader must go back to these articles to understand the population studied. In methods, it is mentioned that plasmids from "clinically important outbreaks" and other species collected in intervening years were added. All this would mean that the sample combined a large number of non-biased BSI isolates (which contain a number of ESBL/carbapenemase producers in contrast with other BSI recent series) with selected isolates, right? These previous works comprise some carbapenemase producers but not this work, why? Fig S1 only specifies the numbers but epidemiological characteristics of isolates are missed. A table that explains the features of the isolates analyzed is really necessary to understand the results which have a sound epidemiological component. Clarify.
- **One of the main findings of the authors is the identification of high-risk plasmid groups which are overrepresented in the collection comprising members with and without ARGs.** Early works by Naomi Datta and others in the 1970s and 1980s, (seminal papers in the epidemiology of AMR plasmids) already described plasmids of the same incompatibility groups that those carrying ARGs in collections of strains from the pre-antibiotic era (Hughes VM, Datta N. 1983. Conjugative plasmids in bacteria of the "preantibiotic" era. Nature 302:725–726 and many others by this and other groups). These works (also by many others, see Diane Taylor chapter in the emblematic Plasmid Biology book, any by Ale Carattoli,...and other reviews on AMR plasmids) already highlighted the circulation of very few and abundant plasmids groups associated with AMR. It is surprising that any of these references have not been included or discussed in the paper. Only very very recent papers by Carattoli and de la Cruz (one from each group and punctually mentioned) are included. More surprising, the authors do not correlate the information about replicons they have (FigS9 of PMID: 34479643-ref12)with the results here.
- **Carbapenemase plasmids.** Genes encoding carbapenemases belong to different plasmid groups (F, IncL,...) which may contain a plethora of different ARGs ..or not. The ARGs have different transmission and evolutionary pathways which have recently been described in a PNAS paper (PMID: 32968015). Neither the paper nor the possibility or the contribution of transposable units to the spread of carbapenemase genes (and bla genes is mentioned, see below). See the comment above about ref 11 and 12.
- **Acquisition and transmission pathways of ARGs.** The section " Hybrid assembly reveals

complex nested diversity associated with key AMR genes, significant chromosomal integration of ARGs and presence of multiple copies in different contexts" reveals simply, the epidemiology of ARGs using blaCTX-M-15 as an example. Different acquisitions have been also demonstrated by many different bla genes (especially those associated with ISEcp1 and IS26). Papers by Sally Partridge (she has several about F plasmids and blaCTX-M-15, and about genetic islands of multiresistant genes in F plasmids, all highly related to this issue are missing) It is a pity that Figure S6 and Figure S7 which mainly reflects the context of the plasmid groups 2 and 3 (also reflecting the major circulating F plasmids in Klebsiella and E. coli nowadays and previously described) only serve to address the mosaicism and recombination of different platforms (in fact these composite Multiresistance islands are also emblematic in A/C, P1, W plasmids).

- Chromosomal location of blaCTX-M-15 (Page 8, end of the third paragraph, Discussion section). The interpretation of the high rate of isolates with the chromosomal location of blaCTXM-15 should be revised and supported with references that previously described this issue. First, the result is not "unexpected". The original publication describing the widespread of blaCTX-M-15 in hospitals around the world already reported the chromosomal location in 8/43 (18%) early isolates which had been arbitrarily collected (PMID: 18258110). Afterward, other reports focused on different lineages of E. coli from humans (6/22; 2 in both plasmid and chromosome) (PMID: 24345742) or livestock (PMID: 25074857) highlighted this issue. The highly frequent chromosomal location has also been reported on Salmonella and Klebsiella. Second, the location of blaCTX-M-15 on successful transposable units (ISEcp1 and IS26 are special insertion sequences) able to integrate into highly recombinant F plasmids and chromosomes should also be considered beyond plasmid cost. Note that some of the plasmids containing blaCTXM15 have different TA system and are not easily lost. Also, many isolates have blaCTXM-15 in plasmids and chromosomes.

- The title and the paper refer to the mobilome. Despite transposable elements appearing in some figures, references to IS/transposons are extremely scarce in the text. Also mentioned in previous works.

Other comments.

- The criteria to define "small", "medium" and "large" plasmids (<10kb, >10kb, >100kb) is vague and should be supported by reference (or justified). Note that plasmid groups 1-11, which are the more abundant, are within two size categories (>10kb, >100kb). F plasmids ranged from 90 to->100kb and this division would split plasmid groups with similar characteristics. Also, the range of 10-100kb includes conjugative and nontransferable plasmids belonging to different plasmid categories (PMID: 20805406 among others)

- Figure 2 and 4. How these figures are linked? Do they contain different information??

- Figure 2. What is the meaning of blaCAR (we do not have this in the CARD database)?

- Figure 6. Which are the carbapenemase genes to build this figure? (Global database that serves as a reference). This info is relevant to also considering the promiscuity of these genes beyond plasmids (see PMID: 32968015).

- Figures (2, S2-S4,...). Figures S2-S4 deserve further discussion as well as the clonal background of the isolates studied. Possible duplicate information in references 11 and/or 12 in relation with clonal background. Explain.

Reviewer #2 (Remarks to the Author):

The manuscript analyzes the plasmidome associated with Gram-negative bacteria (primarily E. coli and Klebsiella spp.) from bloodstream infections between 2009 and 2018 in Oxfordshire, UK. Although limited as a general quest by the use of basically just two host genera, the importance of this study relies on the sampling depth of its dataset. Authors closed the genomes of 738 isolates, which included 1,880 plasmids. To

put it into perspective, published studies on the global bacterial plasmidome include datasets just over 5-7 times larger (see, for instance, Acman (2020) and Redondo-Salvo (2020)).

The manuscript is well structured and includes sufficient technical detail and statistical data to make it robust. Authors used Jaccard Index distances to cluster the 1,880-plasmid dataset into 513 "plasmid groups" (for the meaning of the quotation marks, see below). The concordance of the resulting "plasmid groups" with other plasmid typing methods (NMI = 0.81 with Inc/Replication, NMI = 0.93 with Relaxase Class and NMI = 0.81 with COPLA/PTU) is reassuring, which gives greater validity to the approach. Regrettably, the reference sets are not fully used in further descriptions and discussions, and this should be amended in the revised version of the work. When analyzing the distribution of antimicrobial resistance genes (ARGs) in "plasmid groups" with at least 10 members, authors do not find a single group in which all members carry the same ARG, even if the plasmid components are highly similar, which probably means that ARGs are not relevant for the existence/definition of a "plasmid group". As an example, they showcase blaCTX-M-15 to manifest the diversity of genomic locations for an ARG. Finally, and perhaps the result with more potentially interesting applications, the authors found a reduced list of genes which can be used to predict the risk of an ARG-negative plasmid to acquire an ARG gene.

Below are the comments and suggestions that we hope will help to improve this otherwise sound manuscript.

1. The use of "plasmid" "plasmid group" or "plasmid species" is ill-defined through the text. But it is important that the reader knows exactly what kind of unit the authors are referring to. I will just point out how "diffuse" is this naming by underlying these confusing terms the second part of the Abstract: "We demonstrate that *plasmids* are largely, but not entirely, constrained to host species, although there is substantial overlap between *species of plasmid* gene-repertoire. Most ARGs are carried by a relatively small number of *plasmid groups* with biological features that are predictable. *Plasmids* carrying ARGs (including those encoding carbapenemases) *share a putative 'backbone'* of core genes with those carrying no such genes. These findings suggest that future surveillance should, in addition to tracking *plasmids* currently associated with clinically important genes, focus on identifying and monitoring the dissemination of *high-risk plasmid groups* with the potential to rapidly acquire and disseminate these genes".

This naming imprecision is even more appalling, since the authors did type all the plasmids in their dataset using standard methods (such as Inc/Replicon, MOB and PTU). I do not understand why these data are hardly used in the text. Including this information when describing the "plasmid groups" in, for example, the first paragraph of page 6 would help a wide cross-section of readers to recognize and recall the biological characteristics of the mentioned "plasmid groups" into context. Thus, I would strongly recommend to assign PTUs and/or incompatibility groups to the main clusters used by the authors through the text, by including them in the axis of Fig3 and, most important, in the graph of Figure 4.

2. We would also suggest adding a discussion as to why the sparsifying threshold resulting from the Oxfordshire (BSI) network and the threshold used by Acman et al. for their global dataset are so different (i.e., their JI distance = 0.551 vs. Acman's JI = 0.3). As a consequence, in page 5, authors indicate that 67% of "plasmid groups" (comprising one third of the total n° of plasmids) are unique to the BSI dataset. However, they having used a more stringent JI threshold, could be the cause for an overestimation of the number of unknown "plasmid groups" in the Acman dataset.

3. Finally, a discussion on how their "plasmid groups" relate to the PTUs reported by Redondo-salvo et al (2020) can lead to interesting hindsights on how "plasmid groups" relate to each other. Our expectation is that the authors "plasmid groups" are subsets of

PTUs, and can therefore represent local, temporal or ecological adaptations of given PTUs. See also Coluzzi et al. (2022) MBE 39(6):msac115.

Minor comments:

- First paragraph of Results. The sentence "and 3% (24/168) other Enterobacterales" should read "and 3% (24/738) other Enterobacterales".
- Please, use different colors for panels A and B in Figure 1.
- Second page, last paragraph: "only 4% (22/5143) plasmid groups were" should be "only 4% (22/513) plasmid groups were".
- The method for the pangenome analysis does not explicitly state the % identity used to define shared genes. I'm guessing it is 70%, the Panaroo default. Please, specify.
- In Figure S4, host abundance is used to classify plasmid groups as E. coli or Klebsiella groups. However, the differences in host composition of the dataset (there are 3 times more E. coli hosts than Klebsiella) could have an impact on the results. Could it be explored?
- The Panaroo (25) and Flanker (28) references refer to old versions. Please, update.

Reviewer #3 (Remarks to the Author):

What are the noteworthy results?

The authors present an exciting analysis of a longitudinal BSI dataset and the carriage of AMR genes in different plasmid context through horizontal (inter and intra-species) and vertical transmission. This dataset is notable in terms of the size of complete genomes analyzed from a defined geographic area and provides two primary sampling time points in addition to contextual isolates. It is noteworthy, that 50% of the BSI isolate plasmids contained at least one resistance gene, demonstrating the significance of AMR in clinical settings. The network-based plasmid clustering approach utilized in the study highlighted a large degree of plasmid diversity with many plasmids being unique to a single strain but there is a small number of abundant "types". The observed species level restriction that 86% of the 164 plasmid groups is interesting despite there being members which are mobilizable as well as conjugative. It will be interesting to see this work extended to other Enterobacteriaceae to see if there is just limited flow between Klebsiella and E. coli or if this is a more generalizable trend. The extensive gene sharing between chromosome and plasmid sequences highlights the importance of long-read sequencing for precise placement of these genes in their proper context.

Will the work be of significance to the field and related fields? How does it compare to the established literature? If the work is not original, please provide relevant references.

The work and dataset produced here will be of interest by many in public health studying AMR transmission. The high diversity of plasmids observed in the dataset provides a useful guidance on the extent of sampling required to capture a sufficient degree of plasmid diversity from a population. The rates of plasmid typeability based on replicon and relaxase and restriction of plasmid types to single species is in line with previous work <https://www.ncbi.nlm.nih.gov/pmc/articles/PMC7660255/>. The Network based plasmid clique identification is an alternative method to PTU classification which can provide typing information for epidemiological investigations of plasmids.

Does the work support the conclusions and claims, or is additional evidence needed?

On the whole, the authors present some very interesting results but there are clarity

issues which should be addressed to make sure that their point is coming across clearly.

Are there any flaws in the data analysis, interpretation and conclusions? Do these prohibit publication or require revision?

Overall, I think this is important and high caliber research but I do find parts of the manuscript difficult to follow and there are some inconsistencies that I think would improve understanding of the manuscript. There are multiple issues which I have flagged that make me recommend that revision is needed before this can be accepted.

- The description of the granularity and size of the groupings provided by each of the different approaches [Replicon, Relaxase, PTU] should have a figure or a table and to be discussed in text. I think this provides needed context for the utility of the network based clusters and within the plasmid community the context of the plasmids in terms of replicon/relaxase are frequently what are used. Especially due to the large number of singletons observed

- "Sequence type and host species explained 8% and 7% (Adonis $p=0.001$ for both) of the observed plasmidome variance respectively. ARG content explained a comparatively small amount of variance ($R^2=2\%$, $p=0.001$) as did year of isolation (0.03%, $p=0.005$) and source attribution ($R^2=1.2\%$, $p=0.99$, i.e. suspected focus of infection, only available for a small subset of isolates [198/738])"

- o It is not clear what is being "explained" by these factors or how

- o The biggest explanatory variable may just be frequency of the plasmid group

- o "When we focussed on plasmid groups found in the most common E. coli STs (131/95/73), we observed that most were seen in only a single ST (78/109) but 13 'generalist' groups were seen in all three STs, and accounted for the majority of plasmids (215/400 (54%))."

- o I find the ML component to be a major weakness of the manuscript as there is no context as to what these genes are and what signal is being extracted from them and doesn't connect with the rest of the work being presented. There is no connection to why this ML prediction of "non-AMR" plasmids has value or how it fits into the larger narrative. There are extensive results shown on the plasmid groupings and the gene content analysis which is much more compelling. Most of the plasmids in your dataset are singleton groups (68%) meaning that they lack sufficient similarity to be clustered together, so why is there an expectation that the extracted genetic signal is meaningful? This is been performed on a tiny slice of the data and I have serious doubts regarding its applicability to other datasets. I think this section should be significantly refocused to support the other points made in the manuscript or split off into its own manuscript to explore this specific question in proper detail, as I do think the exploration has merit, but as presented it detracts from the manuscript.

- o

Is the methodology sound? Does the work meet the expected standards in your field?

Yes the methodology is sound for the most part but there are issues with clarity in how they are presented as well as missing details and data, which should be addressed before publication. The network approach is interesting and is executed well but I think that there is a lack of connection to well established typing methods like replicon for which limit the accessibility of the manuscript and this information should be included and highlighted in important cases such as for the blaCTX-m-15 example.

Stylistically, there is extensive use of parenthesis, multiple sentences should be reworked to not require them. They definitely are useful in some situations but they are

pervasive through the manuscript and there are multiple sentences where parentheses are used multiple times.

- Most genes in the pangenomes of common (i.e. containing $n \geq 10$ plasmids) plasmid groups of *E. coli* and *Klebsiella* spp. were non-unique to their group (median % non-unique genes 88%, IQR 67-98%). M

- ($R^2=1.2\%$, $p=0.99$, i.e. suspected focus of infection, only available for a small subset of isolates [198/738])

-

There are also so many parentheses in sections that it hinders readability in part because they are putting different pieces of data together with '/' and it switches from ratios to other groupings. I recommend "/" for ratios and "," for sets

- When we focussed on plasmid groups found in the most common *E. coli* STs (131/95/73), we observed that most were seen in only a single ST (78/109) but 13 'generalist' groups were seen in all three STs, and accounted for the majority of plasmids (215/400 (54%)).

Is there enough detail provided in the methods for the work to be reproduced?

No there is both insufficient detail and missing data for the work to be reproduced.

- I am confused as to how many samples were actually analysed fully as part of this analysis. The authors state that there is "Details of successfully sequenced isolates, those excluded and the stratification and selection methods are available in the appendix" but this is not included in my package. At various points numbers are used but it is not clear if it should be 738 – 215 or if they actually sequenced more isolates and 738 is the true number. The authors need to state plainly in the methods section X samples were sequenced and Y samples have a closed genome and were analysed. Or they need to remove all isolates and mentions which are not fully closed and analysed. It wasn't until I went through Fig S1 that the numbers made sense.
 - o "Incomplete assemblies (where ≥ 1 replicon [i.e. either plasmids or the chromosome] had >1 contig) were excluded from further analysis ($n=215$)."
 - o (1880 plasmids) to conduct the largest, most unselected and comprehensive evaluation of the epidemiology and function of plasmids associated with Gram-negative isolates causing bloodstream infections to date. M"

- The authors only deposited the Illumina raw data into NCBI, they need to deposit the Nanopore data as well. Since they are using closed genomes, it is also important for them to deposit the finalized assemblies for their complete genomes into NCBI for both reproducibility and the benefit of the community as a whole. This is a highly useful dataset and all of the sequence data should be made public.

- Versions of software is inconsistently included, all software should have the version used. I.e. Unicycler, Fly, Ratatosk, Shovill and many others are missing versions of software

- Parameters used for each of the software is also missing and should be included

- Figure 1B – It is unclear what a plasmid "group" entails. How is MDR distinguished from AMR group. How are "no-amr" plasmids grouped? If this is purely based on size of the plasmids, then I have significant concerns about this analysis as there is insufficient detail to understand it properly. If they are based on the PTU's described or other grouping described later then that is reasonable but the authors need to make their units here clear and in the text.

- The description of the granularity and size of the groupings provided by each of the different approaches [Replicon, Relaxase, PTU] should have a figure or a table and to be discussed in text. I think this provides needed context for the utility of the network based clusters and within the plasmid community the context of the plasmids in terms of

replicon/relaxase are frequently what are used.

- I think many people would be interested in seeing the aggregated results that the authors produced as part of the resistance genes and plasmid groups they identified. A supplemental table with the plasmid group, frequency in different ST, frequency of carriage of a resistance gene would be highly useful. The data the authors has produced is complex and has multiple dimensions and so I think re-analysis will be of interest ie.

Group_id,total_members,count_2009,count_2018,ave_size,count_E.coli,count_ST_1,..S Tn,count_klebsiella, count_ST_1,..STn,count_ctx-m-15...

- "We found strong evidence that the plasmidome of BSI isolates was structured by host phylogeny, although there was also vast and persistent background diversity. "
 - o This statement needs to be tied to a specific finding and re-worded to make the meaning more clear. If only 32% (164) of your plasmid groups have >1 member then by definition 68% will be "structure" taxonomically because they only occur once. I found this entire section to be quite difficult to follow.

Minor points:

Figure 2 is referenced after figure 3, these should be re-numbered in text

How was copy number determined? "The 439 plasmids carrying at least one ARG were predominantly large ($\geq 100,000$ bp, 277/439, 63%), low copy number (median 1.80 IQR 1.63-2.37) and conjugative (347/439, 79%). "

MOB-suite and mobsuite occur in the paper

The authors need to be explicit in which versions of kits were used for Nanopore sequencing as there are multiple iterations of the rapid kit. The "native" barcoding kit would have been used in combination with a ligation sequencing kit, these details should be described.

The authors should state that they generated Illumina 150bp reads as described previously.

The authors write on page 2 that "7% (128/1880) were not typable by any method." My suggestion would be to clarify this text to be "any of these tested approaches", since you then go on to describe an alternative graph-based approach. Also, MOB-suite has a universal whole sequence-based plasmid nomenclature implemented <https://www.ncbi.nlm.nih.gov/pmc/articles/PMC7660255/> in addition to an expanded replicon database but the authors limited its use to just relaxase typing.

Mash is probabilistic and so the seed value is important for reproducibility. There can be differences in the mash distances with different seeds and this could have significant impacts on the community detection. I am not sure the degree of sensitivity of their algorithm to changes in the kmer distances. So I will put this as a point for the authors to consider.

"consistent with the hypothesis that the persistence of plasmids is linked to their host range potential." Needs a reference

Supplemental Figures, need to fix multiple plots to include axis labels

Reviewer #4 (Remarks to the Author):

Regarding the manuscript by Samuel Lipworth titled 'The mobilome associated with Gram-negative bloodstream infections: A large-scale observational hybrid sequencing based study' describing data generated from a large set of Enterobacteriales from blood stream infections (BSI) occurring between 2009 and 2018 in Oxfordshire. The manuscript describes a massive sequencing effort to study the plasmidome of clinical samples from the same geographical area. The rationale for such a study is well explained, but especially the methodological approach to validate and compare such a large amount of plasmids is opening for some questions, which needs to be addressed in order to make the analysis and conclusions understandable for an audience within the plasmid epidemiology community. Below, the main concerns is listed.

A) Selection only isolates from BSI seems as a rather biased approach. Why this collection apart from perhaps convenient sampling and the availability of Illumina sequencing? Does this selection represent E. coli from other reservoirs such as the urinary tract or the gut?

B) It is not very clear how many of the included isolates could actually be part of various clonal outbreaks. For instance, how many K. pneumoniae ST490 (suspected hospital clone mentioned in the manuscript) were included in the study? Obviously, including many highly related isolates would give a bias to the study.

C) Please give a detailed quantitative measure of the quality of the hybrid complete plasmid assemblies generated from the 611 strains harboring plasmids. Hybrid assemblies are infamous for creating chimeric assemblies, which should be identified and omitted from the analysis. Was this taken into consideration? Otherwise, how can the authors be sure that plasmid assemblies included in the analysis and comparisons represents the true version of the molecule?

D) Regarding the exclusion criteria of assembled plasmids, can the authors elaborate on these criteria? Was only circular sequences included in the final comparison? What about presence of linear plasmids, which has at least been described in Salmonella and Klebsiella in the literature? Also, was circular sequences with more than one replicon included in the analysis? These could be miss-assemblies but may also be correct, as especially IncF and IncHI plasmids can carry more than one replicon.

E) What %ID cut-off was used for detecting replicons in PlasmidFinder and MOB genes in MOB-Suite? The choice would have had consequences for correct detection of plasmids as e.g. PlasmidFinder is designed to have a cut-off at 80% to detect similar (small) plasmids. Was this cut-off used in the initial analysis before the tool was discarded?

F) On what basis was plasmids categorized into small (>10kb), medium (>10kb; <100kb) and large plasmids (>100 kb)? These numbers seems rather arbitrary. It may have been more relevant to group according to other criteria than size. Or is the division into these three groups based on a size distribution of all the identified plasmids and thus represents the 3 most meaningful peaks in such a distribution analysis?

G) As neither replicon nor MOB typing was helpful to group plasmids, a new method described in the manuscript by the sentence; 'Subsequently, we therefore opted to use a previously described classification approach, utilizing a graph-based Louvain community detection algorithm(10)'. It is not clear from the Materials section, exactly how good this method was to group the different plasmids into meaningful groups/clusters. Please explain in more details, how closely related two plasmids should be in order to be clustered together? What if e.g. a 20% deletion occurred by a single genetic event in a plasmid. Would the original plasmid and the plasmid with the deletion both be put into the same clusters? How was this approach validated to ensure that the clustering made biological sense?

H) The findings concerning the plasmid clustering given in the manuscript is difficult to translate to existing plasmid typing nomenclature. To make the findings presented in the manuscript more applicable for others, some sort of connection between the (unnamed clusters) and replicon-and MOB-types should be considered.

Reviewer #1 (Remarks to the Author):

The manuscript by Lipworth et al provides an apparent exhaustive analysis of the plasmid sequences from isolates of Enterobacterales caused by bloodstream infections (BSI) in a single hospital. The isolates were collected in a timespan of 10 years but most of them correspond to 2009 and 2018. The amount of sequencing and bioinformatic analysis performed is impressive but the novelty of the findings is very limited, and the interpretation of the results deserves a deep and careful revision. All the results that the authors show as “novel” (eg. the notion of high-risk plasmids) or as unexpected (e.g. high rate of the chromosomal location of bla_{CTX-M-15}) are well-known by people working in the field of epidemiology of bacterial antimicrobial resistance. The lack of categorization of the plasmids according to traditionally typing systems (replication, mobilization) and the weak connection between figures and some sections of the paper make it impossible to understand what the results really mean beyond the description of different predominant plasmid groups which may or not carry ARGs (what is the novelty of this?). Despite the amazing and huge available information about some plasmid families and the transmission pathways of bla genes -which is relevant to interpreting and discussing what is explained in the paper-, most of these key publications are missing. This made some statements appear as "novel" or "surprising" when they are not. Finally, it would be appropriate what this publication adds to the previous ones by the authors (ref 11 and 12). Some results could be redundant (clonal background, plasmid diversity).

Response: We thank the reviewer for their overall perspective and thoughts on the manuscript. We have sought to address these concerns as part of the point-by-point response to the comments below. We would argue that although the association of AMR genes with specific plasmid replicon types is well-known, our work highlights novel aspects of plasmid diversity, and the context of AMR genes within this, that can only be ascertained with large-scale, systematic, dense sampling, and the use of reference-free based methods for plasmid classification.

Similarly, although many researchers in the field appreciate that chromosomal integration of AMR genes such as bla_{CTX-M-15} is significant and important, we are not aware of any data that have sought to quantify this by using such systematically and consecutively sampled, and large, isolate collections. Robustly characterising the genetic location (i.e. either chromosomal or plasmid) of bla_{CTX-M-15} without hybrid assembly can be challenging, because the associated flanking sequences are often present in multiple copies within isolates, and any assemblies generated from short-read data typically fragment in such cases. Whilst it is possible with laboratory techniques referred to in the papers the reviewer cites, these are difficult to perform at scale and do not provide detailed information about genetic flanking context such as we describe here.

Traditional typing systems for plasmids such as replicon typing are a very useful reference point but have now been shown in multiple to studies to represent a pretty crude relatedness measure (in the same way as seven-locus MLST is for bacterial lineages), with significant genetic distance observed amongst members of the same replicon group. They also fail to describe plasmids that remain untyped by these established schemes.

The reviewer has asked us to justify the novelty of our work in the context of other important previous publications by others and our own work. We have sought to improve on the context provided, but an exhaustive review is beyond the scope of this manuscript.

Some comments are given for the author's consideration.

Major comments.

- The sample. The criteria to select the isolates should be described to properly interpret the results. The authors have previously published two papers (references 11, 12 which support the 2009 and 2018 collections of isolates) but the reader must go back to these articles to understand the population studied.

Response: We have added a sentence to make the sampling strategy used more immediately understandable without having to refer to our other papers:

“We have previously reported analyses of short-read sequencing data from *E. coli* and *Klebsiella* spp. bloodstream infection isolates in Oxfordshire between 2009 and 2018 (18, 19). In this earlier study we sequenced all available isolates in this time period (i.e. a non-biased, sequential and near complete dataset; n=3468 isolates) with de-duplication to 90 days per patient.”

We have also added a line to the supplementary methods to describe the geography of our area:

“The Oxford University Hospital NHS Foundation Trust microbiology laboratory provides a service for four general/specialist referral hospitals as well as all GP practices (community/primary care) in the region (total catchment population ~805,000).”

In methods, it is mentioned that plasmids from "clinically important outbreaks" and other species collected in intervening years were added. All this would mean that the sample combined a large number of non-biased BSI isolates (which contain a number of ESBL/carbapenemase producers in contrast with other BSI recent series) with selected isolates, right? These previous works comprise some carbapenemase producers but not this work, why?

Response: The reviewer is correct that the sample contained a large number of non-biased BSI isolates and this is one of the unique features about this study that distinguishes it from earlier studies i.e. that the sampling frame is collected in a manner that is not pre-selected based on AMR phenotype. This enables us to systematically evaluate background plasmid diversity without solely focusing on cases where specific AMR genes are present – as has been the focus of almost all of the research effort that the reviewer has referenced. We would argue that a proper, unbiased understanding of the background genetic epidemiology of plasmids is a key component to evaluating AMR gene dissemination in context across these genetic backgrounds.

In addition, as we have described previously (<https://doi.org/10.1186/s13073-021-00947-2>), carbapenemase producers are very rare in *E. coli*/*Klebsiella* spp. BSIs in Oxfordshire (e.g. 1/3461 *E. coli* and 3/556 *Klebsiella* spp.). The epidemiology of plasmids carrying carbapenemase-encoding genes has been extensively investigated elsewhere (e.g. <https://pubmed.ncbi.nlm.nih.gov/32968015/>, <https://pubmed.ncbi.nlm.nih.gov/32094139/>); given our low prevalence locally we did not focus on analysing in detail the small number of carbapenemase-producing plasmids that were identified locally.

Fig S1 only specifies the numbers but epidemiological characteristics of isolates are missed. A table that explains the features of the isolates analyzed is really necessary to understand the results which have a sound epidemiological component. Clarify.

Response: We are not clear from the comment provided what epidemiological details the reviewer would like us to specifically reference, but any details that are not available are not given because they were not used in any of the analysis presented in the study. However we agree that some additional information may help some readers to understand the context of the dataset; this is summarised in a new supplementary table S1 (“Basic epidemiological parameters for isolates included in the study”).

- One of the main findings of the authors is the identification of high-risk plasmid groups which are overrepresented in the collection comprising members with and without ARGs. Early works by Naomi Datta and others in the 1970s and 1980s, (seminal papers in the epidemiology of AMR plasmids) already described plasmids of the same incompatibility groups that those carrying ARGs in collections of strains from the pre-antibiotic era (Hughes VM, Datta N. 1983. Conjugative plasmids in bacteria of the “preantibiotic” era. *Nature* 302:725–726 and many others by this and other groups).

These works (also by many others, see Diane Taylor chapter in the emblematic Plasmid Biology book, any by Ale Carattoli,...and other reviews on AMR plasmids) already highlighted the circulation of very few and abundant plasmids groups associated with AMR. It is surprising that any of these references have not been included or discussed in the paper. Only very very recent papers by Carattoli and de la Cruz (one from each group and punctually mentioned) are included. More surprising, the authors do not correlate the information about replicons they have (FigS9 of PMID: 34479643-ref12) with the results here.

Response: We completely agree with the reviewer that the researchers they have mentioned have written seminal works on AMR plasmid biology/epidemiology – there are many important researchers working in this field. We believe however that we have significantly extended and shed new light on some of these principles and hypotheses that have been previously put forward with this study, and the aim of our study is not to provide an exhaustive review of all the work that has been done previously.

The Datta/Hughes papers referenced by the reviewer used the Murray collection, a set of ~680 Enterobacteriaceae isolates collected over 40 years between 1917-1954 (>70 years ago) across geography, with “many duplicates” (<https://pubmed.ncbi.nlm.nih.gov/6835408/>), “associated metadata [which] is incomplete and somewhat imperfect”, and a “time signature...perhaps reflecting E.G.D. Murray’s changing research interests over time” (<https://pubmed.ncbi.nlm.nih.gov/26411565/>). Although this collection is clearly unique and important, it is of a different era, and represents the kind of sparseness in sampling over geography and time (and collection biases) that we have attempted to avoid in our study. In “Conjugative plasmids in bacteria of the 'pre-antibiotic' era" Hughes and Datta are more particularly interested in conjugative capacity and the presence of plasmids in isolates before versus after the introduction of antibiotics; this is different to our study, in which we are summarising the detailed genetic diversity of AMR and non-AMR plasmids shared across species in geography and recent time. In their second paper published in Nature that year, analysing the same historic collection of isolates (1917-1954), Datta and Hughes focus on understanding the genetic relatedness of historic non-AMR plasmids versus more recent ones that do carry AMR genes – again this is a different context and question to our study, in which we are analysing this genetic overlap contemporaneously. Similarly, a major focus of Prof Carattoli’s work and reviews has been specifically on AMR-associated plasmids; we are keen to characterise wider plasmid diversity – including both AMR-associated plasmids and those without AMR genes – to enable an understanding of the relative risk posed by specific plasmid groups as vectors and potential vectors of AMR genes.

Furthermore, the use of WGS is now well-recognised to provide superior resolution to incompatibility typing and other methods used in earlier studies as plasmids of the same incompatibility groups can be highly genetically diverse. Recent studies demonstrate that accessory gene contents can vary rapidly in relation to more well-conserved genes (e.g., PMID: 35639760 <https://doi.org/10.1101/2022.05.06.490774>), meaning that plasmids with similar replicons can carry vastly different accessory functions. By employing a *k*-mer-based method for evaluating plasmid similarity, we account for both the well-conserved plasmid “backbones” and the accessory gene cargo. As an example from our submitted study, the median genetic distance (which in our paper is 1-Jaccard Index) between isolates with the same PlasmidFinder replicon/Inc type profile (for the most common profiles observed >-20 times in our dataset) is 0.908 (IQR 0.428 - 0.983, where 0 indicates identical kmer profile in the mash sketch, and 1.0 complete divergence). Plasmids carrying an IncF-like replicon belong to 143 genetically distinct plasmid groups in our analysis and those carrying the IncFII_1_pKP91-like replicon belong to 23 plasmid groups. This is consistent with recent studies which demonstrate the limited resolution of traditional plasmid typing schemes (e.g., PMID: 28232822), particularly for plasmids carrying an IncF replicon (e.g., PMID: 32681114, PMID: 33649550), and is consistent with findings from other plasmid-network approaches in the literature (e.g., PMID: 32415210, PMID: 30080134). Additionally, it has been noted that clinically relevant ARG-carrying plasmids can have un-typeable replicons (e.g., PMID: 29370371).

To improve the description of the context of our study as suggested by the reviewer, we have referenced several additional studies and included the following statement:

“Earlier works have demonstrated the similarity of ARG-associated plasmids from the pre- and post-antibiotic era (7-9), hinting that these genes are disseminated amongst well-conserved pre-existing plasmid families.”

We also agree that it is helpful to place our findings in the context of widely used relatedness metrics such as incompatibility groups (as well as the more recently described Plasmid Taxonomic Units, PTUs) to aid interpretation and we have increased the references to these throughout the text as well as adding these to the annotations of the largest plasmid groups identified in our dataset (Figure 4).

- Carbapenemase plasmids. Genes encoding carbapenemases belong to different plasmid groups (F, IncL,...) which may contain a plethora of different ARGs ..or not. The ARGs have different transmission and evolutionary pathways which have recently been described in a PNAS paper (PMID: 32968015). Neither the paper nor the possibility or the contribution of transposable units to the spread of carbapenemase genes (and bla genes is mentioned, see below). See the comment above about ref 11 and 12.

Response: We have in fact already cited this PNAS paper in the submitted manuscript (reference 3) and agree it makes an important contribution to the literature. However, as mentioned previously, this work by David et al focusses only on plasmids which contain four carbapenemase genes. In contrast, our study highlights the extent to which highly genetically related plasmids without these carbapenemase genes are currently circulating - pointing to an intertwined ecology and ongoing exchange of genetic material e.g., by transposons. Carbapenemase-negative plasmids that are highly genetically related to those known to carry major carbapenemase genes represent a major risk for horizontal acquisition of these genes in settings where carbapenemases currently remain a relatively limited clinical problem, such as Oxfordshire.

- Acquisition and transmission pathways of ARGs. The section “ Hybrid assembly reveals complex nested diversity associated with key AMR genes, significant chromosomal integration of ARGs and presence of multiple copies in different contexts” reveals simply, the epidemiology of ARGs using blaCTX-M-15 as an example. Different acquisitions have been also demonstrated by many different bla genes (especially those associated with ISEcp1 and IS26). Papers by Sally Partridge (she has several about F plasmids and blaCTX-M-15, and about genetic islands of multiresistant genes in F plasmids, all highly related to this issue are missing) It is a pity that Figure S6 and Figure S7 which mainly reflects the context of the plasmid groups 2 and 3 (also reflecting the major circulating F plasmids in Klebsiella and E. coli nowadays and previously described) only serve to address the mosaicism and recombination of different platforms (in fact these composite Multiresistance islands are also emblematic in A/C, P1, W plasmids).

Response: We absolutely agree that Sally Partridge is a leader in the field of mobile genetic elements including several papers on *bla*_{CTX-M-15}, however these have either been PCR-based studies (and thus not able to fully characterise the genomic context)^{5,6}, based on small numbers of arbitrarily selected isolates^{7,8} or a series of (excellent) reviews^{9,10}. To our knowledge however neither Prof Partridge, nor anyone else, have systematically studied the genomic context of *bla*_{CTX-M-15} using hybrid sequencing at the same scale and with the same systematic sampling as we have done in this manuscript. We firmly believe that describing the epidemiology of plasmids carrying genes such as *bla*_{CTX-M-15} in a large, unselected, “real-world” clinical dataset is very different and adds value to papers which focus on the genetics of a small number of selected isolates.

- Chromosomal location of *bla*_{CTX-M-15} (Page 8, end of the third paragraph, Discussion section). The interpretation of the high rate of isolates with the chromosomal location of *bla*_{CTX-M-15} should be revised and supported with references that previously described this issue. First, the result is not “unexpected”. The original publication describing the widespread of *bla*_{CTX-M-15} in hospitals around the world already reported the chromosomal location in 8/43 (18%) early isolates which had been arbitrarily collected (PMID: 18258110). Afterward, other reports focused on different lineages of *E. coli* from humans (6/22; 2 in both plasmid and chromosome) (PMID: 24345742) or livestock (PMID: 25074857) highlighted this issue. The highly frequent chromosomal location has also been reported on *Salmonella* and *Klebsiella*. Second, the location of *bla*_{CTX-M-15} on successful transposable units (ISECp1 and IS26 are special insertion sequences) able to integrate into highly recombinant F plasmids and chromosomes should also be considered beyond plasmid cost. Note that some of the plasmids containing *bla*_{CTX-M-15} have different TA system and are not easily lost. Also, many isolates have *bla*_{CTX-M-15} in plasmids and chromosomes.

Response: We agree that this section could have been worded better. The reviewer is absolutely right that chromosomal integration of *bla*_{CTX-M-15} has already been reported. However, what is novel about our dataset is the availability of unselected isolates across a ten-year period (i.e. the ability to compare isolates from 2009 vs 2018 and show increasing chromosomal integration). Interestingly this also seems to be reflected in the literature, including studies referenced by the reviewer (e.g. 18% in PMID: 18258110 (2000-2006, 27% in PMID: 24345742, 8/16 (50%) in PMID: 34485958 (2014-2016) and 23/41 (56%) in this study. We certainly are not the first to show chromosomal integration but the ability to show evidence of a clear trend of increasing chromosomal integration of a key ARG over time is a significant strength of the dataset collected here and we think highly relevant to ongoing surveillance studies. We have amended this section in the Discussion as follows:

“The high proportion of isolates with chromosomally integrated ARGs (and apparent increase across the study period for *E. coli*) supports findings from previous studies (15,16) and may either represent a success of plasmids in conferring survival benefits to their host while lowering their own associated fitness cost or a success of the host by lowering its dependence on the presence of the plasmid. It may also reflect their frequent association with transposable units (6) (e.g. ISECp1/IS26 for *bla*_{CTX-M-15}) which facilitate their integration and dissemination.”

- The title and the paper refer to the mobilome. Despite transposable elements appearing in some figures, references to IS/transposons are extremely scarce in the text. Also mentioned in previous works.

Response: We agree that the focus is predominantly on plasmids and have amended the title as suggested to “The plasmidome associated with Gram-negative bloodstream infections: A large-scale observational hybrid sequencing based study.”

Other comments.

- The criteria to define “small”, “medium” and “large” plasmids (<10kb, >10kb, >100kb) is vague and should be supported by reference (or justified). Note that plasmid groups 1-11, which are the more abundant, are within two size categories (>10kb, >100kb). F plasmids ranged from 90 to >100kb and this division would split plasmid groups with similar characteristics. Also, the range of 10-100kb includes conjugative and nontransferable plasmids belonging to different plasmid categories (PMID: 20805406 among others)]

Response: We initially split the data in a similar way to other recently published plasmid studies (e.g. <https://www.science.org/doi/10.1126/sciadv.abe3868> where plasmids are split arbitrarily into “small” <=10,000bp and “large” >10,000 bp). However, our exploratory analyses showed that there was an intermediate group of plasmids (e.g. groups 9-17 in Figure 4) which are mostly non-Inc-F and non-Col and less easy to identify using existing typing schemes and so we thought this was an interesting comparator group. These categorisations do not form an important part of our analysis and so will not adversely affect the wider interpretation of our manuscript.

- Figure 2 and 4. How these figures are linked? Do they contain different information??

Response: Figure 3 shows a umap plot of plasmidomes (i.e. all plasmid content for an isolate is considered as a single unit) coloured by the characteristics shown in the legend (species, ARGs, source, year). Figure 4 is a network plot for individual plasmids coloured according to whether or not they carry an ARG. We feel that presenting these hierarchical levels of information in separate plots is useful to help readers understand these related but distinct findings.

- Figure 2. What is the meaning of blaCAR (we do not have this in the CARD database)?

Response: Apologies, this was a typo and should read blaCARB; we have updated.

- Figure 6. Which are the carbapenemase genes to build this figure? (Global database that serves as a reference). This info is relevant to also considering the promiscuity of these genes beyond plasmids (see PMID: 32968015).

Response: The “carbapenemase gene” grouping includes all those variants identified classified as conferring resistance to the class “Carbapenem” in AMRFinder (i.e. mostly *bla*_{IMP}/*bla*_{KPC}/*bla*_{NDM} and relevant *bla*_{OXA} variants). We have now clarified this in the legend of Figure 6.

- Figures (2, S2-S4,...). Figures S2-S4 deserve further discussion as well as the clonal background of the isolates studied. Possible duplicate information in references 11 and/or 12 in relation with clonal background. Explain.

Response: We are pleased that the reviewer finds figures S2-4 interesting but the analysis is rather secondary to the main focus of this manuscript and there is not space to discuss further here (though we agree it possibly deserves further exploration in subsequent work). We feel that further discussion of the clonal background would duplicate our earlier work¹¹ in which this is discussed in a huge amount of detail. We agree however that it is useful to signpost interested readers to this context before they reach the Methods section (especially as this is at the end of the paper in the Nature Comms format). We have therefore added the following to the last paragraph of the discussion:

“Using short read sequencing we have previously described in detail the population dynamics of *E. coli* and *Klebsiella* spp. BSI isolates collected between 2009-2018 in Oxfordshire, UK(12).”

Reviewer #2 (Remarks to the Author):

The manuscript analyzes the plasmidome associated with Gram-negative bacteria (primarily *E. coli* and *Klebsiella* spp.) from bloodstream infections between 2009 and 2018 in Oxfordshire, UK. Although limited as a general quest by the use of basically just two host genera, the importance of this study relies on the sampling depth of its dataset. Authors closed the genomes of 738 isolates, which included 1,880 plasmids. To put it into perspective, published studies on the global bacterial plasmidome include datasets just over 5-7 times larger (see, for instance, Acman (2020) and Redondo-Salvo (2020)).

The manuscript is well structured and includes sufficient technical detail and statistical data to make it robust. Authors used Jaccard Index distances to cluster the 1,880-plasmid dataset into 513 “plasmid groups” (for the meaning of the quotation marks, see below). The concordance of the resulting “plasmid groups” with other plasmid typing methods (NMI = 0.81 with Inc/Replication, NMI = 0.93 with Relaxase Class and NMI = 0.81 with COPLA/PTU) is reassuring, which gives greater validity to the approach. Regrettably, the reference sets are not fully used in further descriptions and discussions, and this should be amended in the revised version of the work. When analyzing the distribution of antimicrobial resistance genes (ARGs) in “plasmid groups” with at least 10 members, authors do not find a single group in which all members carry the same ARG, even if the plasmid components are highly similar, which probably means that ARGs are not relevant for the existence/definition of a “plasmid group”. As an example, they showcase *bla*_{CTX-M-15} to manifest the diversity of genomic locations for an ARG. Finally, and perhaps the result with

more potentially interesting applications, the authors found a reduced list of genes which can be used to predict the risk of an ARG-negative plasmid to acquire an ARG gene.

Below are the comments and suggestions that we hope will help to improve this otherwise sound manuscript.

Response: We are very grateful for the time the reviewer has taken in reviewing our manuscript and are glad they find it of interest.

1. The use of “plasmid” “plasmid group” or “plasmid species” is ill-defined through the text. But it is important that the reader knows exactly what kind of unit the authors are referring to. I will just point out how “diffuse” is this naming by underlying these confusing terms the second part of the Abstract: “We demonstrate that *plasmids* are largely, but not entirely, constrained to host species, although there is substantial overlap between *species of plasmid* gene-repertoire. Most ARGs are carried by a relatively small number of *plasmid groups* with biological features that are predictable. *Plasmids* carrying ARGs (including those encoding carbapenemases) *share a putative ‘backbone’* of core genes with those carrying no such genes. These findings suggest that future surveillance should, in addition to tracking *plasmids currently associated with clinically important genes*, focus on identifying and monitoring the dissemination of *high-risk plasmid groups* with the potential to rapidly acquire and disseminate these genes”.

Response: We have used the term “species” throughout to refer to bacterial species (e.g. *E. coli/Klebsiella pneumoniae*) in accordance with standard taxonomy. There is no concept of “plasmid species” used in our manuscript. We have clarified that we mean bacterial species not plasmid species after the first time this is used:

“Plasmids are thought to facilitate the rapid dissemination of these genes within and between bacterial species.”

The term “plasmid” is used to refer to an individual plasmid; “plasmids” to refer to several plasmids.

We agree that the term “plasmid group” is not currently explicitly defined and we have now done this after it is first introduced:

“Subsequently, we therefore opted to use a previously described classification approach, utilizing a graph-based Louvain community detection algorithm(14) (see methods) which has the advantage of not being reliant on reference databases for group assignment and is thus able to classify all plasmids into groups (hereafter referred to as “plasmid groups”).”

This naming imprecision is even more appalling, since the authors did type all the plasmids in their dataset using standard methods (such as Inc/Replicon, MOB and PTU). I do not

understand why these data are hardly used in the text. Including this information when describing the “plasmid groups” in, for example, the first paragraph of page 6 would help a wide cross-section of readers to recognize and recall the biological characteristics of the mentioned “plasmid groups” into context. Thus, I would strongly recommend to assign PTUs and/or incompatibility groups to the main clusters used by the authors through the text, by including them in the axis of Fig3 and, most important, in the graph of Figure 4.

Response: We agree that this suggestion helps improve contextualise the network-based plasmid groups and have added Inc/Rep and PTU assignments into the graph of Figure 4. We have also added this throughout the text where we feel it aids interpretation. We tried incorporating this into the X-axis of figure 3, however there was insufficient space to make this readable.

2. We would also suggest adding a discussion as to why the sparsifying threshold resulting from the Oxfordshire (BSI) network and the threshold used by Acman et al. for their global dataset are so different (i.e., their JI distance = 0.551 vs. Acman’s JI = 0.3). As a consequence, in page 5, authors indicate that 67% of “plasmid groups” (comprising one third of the total n^0 of plasmids) are unique to the BSI dataset. However, they having used a more stringent JI threshold, could be the cause for an overestimation of the number of unknown “plasmid groups” in the Acman dataset.

Response: The distances we use are 1-JI, thus in our paper a distance of 0 indicates 100% kmer similarity in the mash sketch whereas in the Acman paper 1 indicates 100% similarity. The Acman JI of 0.3 translates to a distance of 0.7 in our study. Nevertheless, the reviewer is correct that our threshold is more stringent than the Acman threshold i.e. 0.551 is closer to 0 than 0.7). This is not unexpected given that Acman et al selected their threshold based on congruence with Inc-typing. Additionally, Acman and colleagues analysed an agnostic and more diverse plasmid dataset pulled from NCBI. A higher diversity of plasmid sequences meant that a less stringent threshold was needed to resolve the plasmids into biologically relevant clusters. In contrast, our plasmids were all sampled from ecologically related *Enterobacteriaceae*, meaning our plasmids were less diverse compared to the Acman dataset, and hence a more stringent threshold was needed to reveal the underlying population structure.

Notably, the thresholds are only used to sparsify the network rather than to define cluster groupings (which is done by the Louvain algorithm). By undertaking a sensitivity analysis using the Acman distance of 0.7 to sparsify the network instead of 0.551 as we have, of the 331 (of 5080 total) plasmid groups containing at least one plasmid from the Oxfordshire dataset, 211/331 (64%) are unique to Oxford (vs 326/484 [64%] in the analysis presented). The choice of threshold therefore does not appear to substantially affect the interpretation of this analysis.

3. Finally, a discussion on how their “plasmid groups” relate to the PTUs reported by Redondo-salvo et al (2020) can lead to interesting hindsights on how “plasmid groups” relate to each other. Our expectation is that the authors “plasmid groups” are subsets of PTUs, and can

therefore represent local, temporal or ecological adaptations of given PTUs. See also Coluzzi et al. (2022) MBE 39(6):msac115.

Response: The reviewer is correct to speculate that (as one would expect) the “plasmid groups” identified in this work correspond to subsets of PTUs. We agree that it is interesting that these subsets of PTUs appear well structured within the phylogeny in Figure 2 (mapping between plasmid groups and PTUs shown in Figure 4); further exploration of whether this is a signal of adaptation would be interesting future work.

We have added the following to the discussion:

“The plasmid groups we identify appear to be subsets of the recently described PTU typing system (e.g. plasmid groups 3/4/6/7/8 in Figure 4 all correspond to PTU-FE) and there appears to be a non-random distribution of these across the phylogeny in Figure 2, possibly indicating that these subsets are adapted to particular sequence types (or local/ecological niches).”

Minor comments:

- First paragraph of Results. The sentence “and 3% (24/168) other Enterobacterales” should read “and 3% (24/738) other Enterobacterales”.

Response: Thank you for spotting this error – we have corrected.

- Please, use different colors for panels A and B in Figure 1.

Response: We agree this was confusing and have altered in line with the reviewer’s suggestion.

- Second page, last paragraph: “only 4% (22/5143) plasmid groups were” should be “only 4% (22/513) plasmid groups were”.

Response: Thank you for spotting this typo - we have corrected.

- The method for the pangenome analysis does not explicitly state the % identity used to define shared genes. I’m guessing it is 70%, the Panaroo default. Please, specify.

Response: Indeed - panaroo was run using the default 0.7 --family_threshold setting, we have clarified this:

“Pangenomes were analysed using Panaroo (v1.2.8 --clean-mode sensitive --family_threshold 0.7)”

- In Figure S4, host abundance is used to classify plasmid groups as *E. coli* or *Klebsiella* groups. However, the differences in host composition of the dataset (there are 3 times more *E. coli* hosts than *Klebsiella*) could have an impact on the results. Could it be explored?

Response: We agree that it would be theoretically possible to perform some sort of sampling in order to balance the datasets, however the dataset we have reflects the imbalance in the isolation of these organisms that is seen in bloodstream infections in the real world (i.e. *E. coli* is

far more common than *Klebsiella* spp.). It is unlikely that rebalancing these datasets would alter the interpretation of clusters as belonging predominantly to one or other species given the strong structuring by phylogeny we demonstrate in the rest of the manuscript. The point we seek to illustrate is that even in the context of this imbalanced dataset, there is still substantial overlap between the pangenomes, particularly at the chromosomal level. We have made a note of this as a limitation in the discussion:

“The species imbalance in the dataset reflects the fact that *E. coli* BSIs are more common than those caused by *Klebsiella* spp. but may nevertheless bias inferences about plasmid and gene sharing within and between species.”

- The Panaroo (25) and Flanker (28) references refer to old versions. Please, update.

Response: Thanks for spotting this, we have fixed.

Reviewer #3 (Remarks to the Author):

What are the noteworthy results?

The authors present an exciting analysis of a longitudinal BSI dataset and the carriage of AMR genes in different plasmid context through horizontal (inter and intra-species) and vertical transmission. This dataset is notable in terms of the size of complete genomes analyzed from a defined geographic area and provides two primary sampling time points in addition to contextual isolates. It is noteworthy, that 50% of the BSI isolate plasmids contained at least one resistance gene, demonstrating the significance of AMR in clinical settings. The network-based plasmid clustering approach utilized in the study highlighted a large degree of plasmid diversity with many plasmids being unique to a single strain but there is a small number of abundant “types”. The observed species level restriction that 86% of the 164 plasmid groups is interesting despite there being members which are mobilizable as well as conjugative. It will be interesting to see this work extended to other

Enterobacteriaceae to see if there is just limited flow between *Klebsiella* and *E. coli* or if this is a more generalizable trend. The extensive gene sharing between chromosome and plasmid sequences highlights the importance of long-read sequencing for precise placement of these genes in their proper context.

Response: We are grateful to the reviewer for their time in reading and commenting on the manuscript – we are pleased a number of features are of interest. We agree that extending this to other species and quantifying the extent of sharing as part of future work would be interesting.

Will the work be of significance to the field and related fields? How does it compare to the established literature? If the work is not original, please provide relevant references.

The work and dataset produced here will be of interest by many in public health studying AMR transmission. The high diversity of plasmids observed in the dataset provides a useful guidance on the extent of sampling required to capture a sufficient degree of plasmid diversity from a population. The rates of plasmid typeability based on replicon and relaxase and restriction of plasmid types to single species is in line with previous work <https://www.ncbi.nlm.nih.gov/pmc/articles/PMC7660255/>. The Network based plasmid clique identification is an alternative method to PTU classification which can provide typing information for epidemiological investigations of plasmids.

Does the work support the conclusions and claims, or is additional evidence needed?

On the whole, the authors present some very interesting results but there are clarity issues which should be addressed to make sure that their point is coming across clearly.

Response: We have tried to improve clarity throughout – please see specific response to the other reviewers and to the points below.

Are there any flaws in the data analysis, interpretation and conclusions? Do these prohibit publication or require revision?

Overall, I think this is important and high caliber research but I do find parts of the manuscript difficult to follow and there are some inconsistencies that I think would improve understanding of the manuscript. There are multiple issues which I have flagged that make me recommend that revision is needed before this can be accepted.

- The description of the granularity and size of the groupings provided by each of the different approaches [Replicon, Relaxase, PTU] should have a figure or a table and to be discussed in text. I think this provides needed context for the utility of the network based clusters and within the plasmid community the context of the plasmids in terms of replicon/relaxase are frequently what are used. Especially due to the large number of singletons observed.

Response: Plasmids in the groups we describe are more similar to each other than those in clusters created by the other methods. We agree with the reviewer that it is helpful to discuss and illustrate this point. We have therefore inserted the following into the text as well as providing a new supplementary table (S2) and figure (S2):

“...which has the advantage of not being reliant on reference databases for group assignment and is thus able to classify all plasmids into groups (hereafter referred to as "plasmid groups"). These Louvain-based plasmid groups generally clustered plasmids together at a lower distance threshold (i.e. more similar) than the other methods tests (median: 0.251, IQR: 0.051-0.522 vs 0.692 IQR: 0.561-0.852 for COPLA/PTU clusters, 0.968 IQR 0.856-1.000 Mobsuite/Relaxase clusters, 0.664 IQR (0.367-0.928) Plasmidfinder/Replicon-typing clusters Figure S2)

“...but only 33 (6%) contained ≥ 10 plasmids, and most were singletons (349/513 (68%)). As expected given the more closely related groupings identified by the Louvain-based approach, this method created more groups compared to the others tested and more of these were singletons (Table S2).

- “Sequence type and host species explained 8% and 7% (Adonis $p=0.001$ for both) of the observed plasmidome variance respectively. ARG content explained a comparatively small amount of variance ($R^2=2\%$, $p=0.001$) as did year of isolation (0.03%, $p=0.005$) and source attribution ($R^2=1.2\%$, $p=0.99$, i.e. suspected focus of infection, only available for a small subset of isolates [198/738])”

- o It is not clear what is being “explained” by these factors or how

Response: We apologise, there was a mistake in the caption of Figure 3 which we have now corrected.

“A Umap projection of distances (measured by gene presence/absence) between the plasmidomes of isolates (each point represents the plasmidome, i.e. all plasmid sequences of a single isolate).”

The permanova tests the null hypothesis that there is no difference between the centroids/spread of plasmidomes (as measured by Jaccard distance calculated from gene presence/absence) between groups. In this case therefore the test suggests that there is a difference in plasmidome content for isolates from different STs/species and, to a lesser extent, year of isolation/ARG content. We have clarified this in the text:

“Sequence type and host species explained 8% and 7% (Adonis $p=0.001$ for both) of the observed variance in gene content between plasmidomes respectively.”

We have additionally clarified our definition of plasmidome in the methods:

“Plasmidome pangenomes (where all plasmids carried by an isolate are considered as a single unit) were analysed using Panaroo”

- o The biggest explanatory variable may just be frequency of the plasmid group

Response: Plasmid group is not used in this model in which distances between plasmidomes (all plasmids carried by an isolate considered as a single unit) are calculated as Jaccard distances based on gene presence/absence matrices.

- o “When we focussed on plasmid groups found in the most common E. coli STs (131/95/73), we observed that most were seen in only a single ST (78/109) but 13 ‘generalist’ groups were seen in all three STs, and accounted for the majority of plasmids (215/400 (54%)).”

Response: We apologise if we are misunderstanding the reviewer here but this comment looks incomplete and we are not sure what is being queried?

o I find the ML component to be a major weakness of the manuscript as there is no context as to what these genes are and what signal is being extracted from them and doesn't connect with the rest of the work being presented. There is no connection to why this ML prediction of "non-AMR" plasmids has value or how it fits into the larger narrative. There are extensive results shown on the plasmid groupings and the gene content analysis which is much more compelling. Most of the plasmids in your dataset are singleton groups (68%) meaning that they lack sufficient similarity to be clustered together, so why is there an expectation that the extracted genetic signal is meaningful? This is been performed on a tiny slice of the data and I have serious doubts regarding its applicability to other datasets. I think this section should be significantly refocused to support the other points made in the manuscript or split off into its own manuscript to explore this specific question in proper detail, as I do think the exploration has merit, but as presented it detracts from the manuscript.

Response: The list of genes identified by the pangenome-wide association study as being significantly associated with AMR gene-positive plasmid groups is presented in Table S4. Notably, of those that are annotatable many are insertion sequences/transposons/virulence factors/heavy metal resistance genes/toxin-antitoxin systems – i.e. factors that would biologically be likely to be associated with adaptation/selection/persistence. We disagree that the results presented here are not likely to be applicable to other datasets - we have demonstrated the good generalisability of a model trained using these features on a spatially and temporally independent dataset. We agree that this work could be further developed and refined and we hope the data we present will motivate further such efforts. We note that Reviewer 2 found this to be one of the most interesting parts of the manuscript and would prefer to keep it in this work.

It is not correct to say that this analysis has been performed on a tiny slice of the data. All plasmids and all plasmid groups were included in the pangenome wide association study (which constituted the feature selection part of the analysis). As mentioned in the methods, when evaluating the predictive performance of these features using the Oxfordshire dataset, 1439 plasmids (1880 - 439 (plasmids with AMR genes) - 2 (plasmids in which prokka found no CDS)) were included of which 609 were in AMR gene-associated groups. The reviewer is correct that AMR gene-containing singletons are excluded from the ML training/testing set, but these represent a relatively small number of AMR gene-containing plasmids (because as mentioned, most AMR gene-containing plasmids cluster into a relatively small number of plasmid groups – i.e. 81% of these are found in 8 plasmid groups). If the presence of a high number of singletons in our dataset biased feature selection or resulted in overfitting we would expect our model to generalise poorly with the global holdout dataset - but this was not the case. Whilst it is true that the available testing set (the global dataset) was smaller (due in part to the bias of the existing

literature towards AMR gene-carrying plasmids and poor availability of metadata), we feel that our findings remain robust and of interest.

Is the methodology sound? Does the work meet the expected standards in your field?
Yes the methodology is sound for the most part but there are issues with clarity in how they are presented as well as missing details and data, which should be addressed before publication. The network approach is interesting and is executed well but I think that there is a lack of connection to well established typing methods like replicon for which limit the accessibility of the manuscript and this information should be included and highlighted in important cases such as for the blaCTX-m-15 example.

Response: We agree that better connection to established typing methods is helpful and we have now included this throughout (e.g. Fig.4 and in the bla_{CTX-M-15} section as suggested).

Stylistically, there is extensive use of parenthesis, multiple sentences should be reworked to not require them. They definitely are useful in some situations but they are pervasive through the manuscript and there are multiple sentences where parentheses are used multiple times.

- Most genes in the pangenomes of common (i.e. containing $n \geq 10$ plasmids) plasmid groups of E. coli and Klebsiella spp. were non-unique to their group (median % non-unique genes 88%, IQR 67-98%). M
- ($R^2=1.2\%$, $p=0.99$, i.e. suspected focus of infection, only available for a small subset of isolates [198/738])
-

There are also so many parentheses in sections that it hinders readability in part because they are putting different pieces of data together with '/' and it switches from ratios to other groupings. I recommend "/" for ratios and "," for sets

- When we focussed on plasmid groups found in the most common E. coli STs (131/95/73), we observed that most were seen in only a single ST (78/109) but 13 'generalist' groups were seen in all three STs, and accounted for the majority of plasmids (215/400 (54%)).

Response: We have tried to reduce the use of parentheses throughout and have adopted the stylistic suggestion of the reviewer regarding ratios and groupings. A lot of the instances of our parenthesis use occur when we have given a percentage in order to also give the numbers in the ratio; we are happy to be guided by the stylistic preference of the editors in this regard.

Is there enough detail provided in the methods for the work to be reproduced?

No there is both insufficient detail and missing data for the work to be reproduced.

- I am confused as to how many samples were actually analysed fully as part of this analysis. The authors state that there is "Details of successfully sequenced isolates, those excluded and

the stratification and selection methods are available in the appendix” but this is not included in my package. At various points numbers are used but it is not clear if it should be 738 – 215 or if they actually sequenced more isolates and 738 is the true number. The authors need to state plainly in the methods section X samples were sequenced and Y samples have a closed genome and were analysed. Or they need to remove all isolates and mentions which are not fully closed and analysed. It wasn't until I went through Fig S1 that the numbers made sense.

Response: We have now clarified this in the first sentence of the results and referenced figure S1 earlier:

“We sequenced and assembled n=953 isolates of which n=738 were complete and included in subsequent analysis (Figure S1).”

And later in the first paragraph:

“In total, these 738 isolates carried 1,880 plasmids with a median of 2 plasmids per isolate”

We have also clarified the methods:

“A breakdown of successfully sequenced isolates and those excluded is shown in Figure S1 and the stratification and selection methods are detailed in the appendix.”

o “Incomplete assemblies (where ≥ 1 replicon [i.e. either plasmids or the chromosome] had >1 contig) were excluded from further analysis (n=215).”

o (1880 plasmids) to conduct the largest, most unselected and comprehensive evaluation of the epidemiology and function of plasmids associated with Gram-negative isolates causing bloodstream infections to date. M”

Response: We hope the above clarifies this.

- The authors only deposited the Illumina raw data into NCBI, they need to deposit the Nanopore data as well. Since they are using closed genomes, it is also important for them to deposit the finalized assemblies for their complete genomes into NCBI for both reproducibility and the benefit of the community as a whole. This is a highly useful dataset and all of the sequence data should be made public.

Response: We agree that open data sharing is critically important and guarantee that this data will be available on a public repository by the time of publication and apologise that it is not yet available. This is not deliberate (the lead author has been in full time clinical service since this was submitted) and we will ensure that the reads and assemblies are available as soon as possible.

- Versions of software is inconsistently included, all software should have the version used. I.e. Unicycler, Fly, Ratatosk, Shovill and many others are missing versions of software

Response: We have included all versions of software now and apologise for this omission in the original submission.

- Parameters used for each of the software is also missing and should be included.

Response: All software was run using default settings except where otherwise stated - we have clarified this:

“All bioinformatic programmes were run using default settings unless otherwise specified”.

- Figure 1B – It is unclear what a plasmid “group” entails. How is MDR distinguished from AMR group. How are “no-amr” plasmids grouped? If this is purely based on size of the plasmids, then I have significant concerns about this analysis as there is insufficient detail to understand it properly. If they are based on the PTU’s described or other grouping described later then that is reasonable but the authors need to make their units here clear and in the text.

Response: We have updated this legend to clarify that plasmid groups are defined using the Louvain-based method. As the reviewer rightly points out, this was confusing because the graph-based plasmid groups are used before the idea is introduced. We have also therefore tried to make this clearer in the text:

“Rarefaction analysis suggested that a substantial number of plasmid groups (defined using a graph-based clustering method, see below)”

No-AMR plasmids contain 0 ARGs - this is nothing to do with plasmid size and refers to part A of this figure, we hope that by adjusting this colour scheme it is now clearer what this legend is referring to.

- The description of the granularity and size of the groupings provided by each of the different approaches [Replicon, Relaxase, PTU] should have a figure or a table and to be discussed in text. I think this provides needed context for the utility of the network based clusters and within the plasmid community the context of the plasmids in terms of replicon/relaxase are frequently what are used.

Response: We agree and have adopted this suggestion in Figure S2 and Table S2.

- I think many people would be interested in seeing the aggregated results that the authors produced as part of the resistance genes and plasmid groups they identified. A supplemental table with the plasmid group, frequency in different ST, frequency of carriage of a resistance gene would be highly useful. The data the authors has produced is complex and has multiple dimensions and so I think re-analysis will be of interest

ie.

Group_id,total_members,count_2009,count_2018,ave_size,count_E.coli,count_ST_1,..STn,count_klebsiella, count_ST_1,..STn,count_ctx-m-15...

Response: Rather than summarising this data as suggested, we have uploaded the raw output from our analysis giving all of this information for every plasmid as a new supplementary data file. We think this is more useful to readers who wish to re-analyse the data than a summary table.

- “We found strong evidence that the plasmidome of BSI isolates was structured by host phylogeny, although there was also vast and persistent background diversity.”
 - o This statement needs to be tied to a specific finding and re-worded to make the meaning more clear. If only 32% (164) of your plasmid groups have >1 member then by definition 68% will be “structure” taxonomically because they only occur once. I found this entire section to be quite difficult to follow.

Response: As discussed above, this section is not linked to plasmid groups but to gene content across all plasmids within isolates (i.e. the plasmidome; this distance is derived from the Panaroo gene presence/absence matrix). As above we have clarified this in the text and methods and have additionally clarified what we mean in this sentence and have reordered it so that it is more obvious which part of our analysis it refers to:

“We found strong evidence that the pangenome of the plasmidome of BSI isolates was structured by host phylogeny, although there was also vast and persistent background diversity.”

Minor points:

Figure 2 is referenced after figure 3, these should be re-numbered in text

Response: Thanks for pointing this out, we have amended.

How was copy number determined? “The 439 plasmids carrying at least one ARG were predominantly large ($\geq 100,000$ bp, 277/439, 63%), low copy number (median 1.80 IQR 1.63-2.37) and conjugative (347/439, 79%). “

Response: We have clarified that:

“Unicycler was used to estimate plasmid copy number.”

MOB-suite and mobsuite occur in the paper

Response: Thanks - we now only use MOB-suite which is what the package developers use.

The authors need to be explicit in which versions of kits were used for Nanopore sequencing as there are multiple iterations of the rapid kit. The “native” barcoding kit would have been used in combination with a ligation sequencing kit, these details should be described.

Response: We have added this detail.

The authors should state that they generated Illumina 150bp reads as described previously.

Response: We have added this as suggested.

The authors write on page 2 that “7% (128/1880) were not typable by any method.” My suggestion would be to clarify this text to be “any of these tested approaches”, since you then go on to describe an alternative graph-based approach. Also, MOB-suite has a universal whole sequence-based plasmid nomenclature implemented <https://www.ncbi.nlm.nih.gov/pmc/articles/PMC7660255/> in addition to an expanded replicon database but the authors limited its use to just relaxase typing.

Response: We agree with this comment and have added the caveat suggested.

Mash is probabilistic and so the seed value is important for reproducibility. There can be differences in the mash distances with different seeds and this could have significant impacts on the community detection. I am not sure the degree of sensitivity of their algorithm to changes in the kmer distances. So I will put this as a point for the authors to consider.

Response: We used mash on default setting (and so the seed is 0). As above we have inserted a sentence to make the use of default settings unless otherwise mentioned explicit.

“consistent with the hypothesis that the persistence of plasmids is linked to their host range potential.” Needs a reference

Response: We have modified this to:

“suggesting that the persistence of plasmids may be linked to their host range potential.”

Supplemental Figures, need to fix multiple plots to include axis labels

Response: We have fixed as suggested.

Reviewer #4 (Remarks to the Author):

Regarding the manuscript by Samuel Lipworth titled ‘The mobilome associated with Gram-negative bloodstream infections: A large-scale observational hybrid sequencing based study’ describing data generated from a large set of Enterobacteriales from blood stream infections (BSI) occurring between 2009 and 2018 in Oxfordshire. The manuscript describes a massive

sequencing effort to study the plasmidome of clinical samples from the same geographical area. The rationale for such a study is well explained, but especially the methodological approach to validate and compare such a large amount of plasmids is opening for some questions, which needs to be addressed in order to make the analysis and conclusions understandable for an audience within the plasmid epidemiology community. Below, the main concerns is listed.

A) Selection only isolates from BSI seems as a rather biased approach. Why this collection apart from perhaps convenient sampling and the availability of Illumina sequencing? Does this selection represent *E. coli* from other reservoirs such as the urinary tract or the gut?

Response: We agree with the reviewer that in an ideal world with unlimited resources performing a similar analysis on e.g. isolates from the gut and urinary tract would be worthwhile. Ultimately there is a substantial resource limitation here as all isolates had to be sequenced on two modalities (indeed the sequencing effort here is far larger than in most existing hybrid sequencing studies). We agree that it will be interesting to compare the data from this study to those from other ecological reservoirs as these become available but we think there is a strong rationale for starting with bloodstream infections. *E. coli/Klebsiella* spp. BSI isolates are almost always clinically relevant and represent the most severe and invasive forms of disease.

B) It is not very clear how many of the included isolates could actually be part of various clonal outbreaks. For instance, how many *K. pneumoniae* ST490 (suspected hospital clone mentioned in the manuscript) were included in the study? Obviously, including many highly related isolates would give a bias to the study.

Response: The breakdown of isolates is shown in Figure S1 - 547/738 (74%) of isolates were selected in an unbiased and sequential manner. Of the 191/738 (26%) isolates selected as part of the enriched set as described in the methods, 17/191 (9%) were *K. pneumoniae* ST 490 and 55/191 (29%) belonged to the predominant *E. coli* clones (STs 131/95/73/69). Whilst it is true that sequencing isolates from dominant clones can bias the data, it is also true that such clones dominate the population structure of Oxfordshire BSI isolates (and global isolates in the case of *E. coli*¹¹⁻¹³).

C) Please give a detailed quantitative measure of the quality of the hybrid complete plasmid assemblies generated from the 611 strains harboring plasmids. Hybrid assemblies are infamous for creating chimeric assemblies, which should be identified and omitted from the analysis. Was this taken into consideration? Otherwise, how can the authors be sure that plasmid assemblies included in the analysis and comparisons represents the true version of the molecule?

Response: Hybrid sequencing represents the current gold standard in the field for complete and accurate assemblies. We are not aware of major problems in creating chimeric assemblies and can find no reference to this in the literature. Clearly no assembly method is perfect, but Unicycler is a very widely used platform in the field which has been shown to outperform other assembly methods even when read depth and accuracy are relatively low¹⁴. We have provided

quality statistics using Quast (new supplementary dataset) however this output is of limited use for hybrid assemblies. We would caution that the output from standard QC workflows is difficult to interpret for hybrid assemblies since the true version of the molecule is not known (but would argue that the hybrid assembly is as close as one can reasonably get to this in the current research environment).

D) Regarding the exclusion criteria of assembled plasmids, can the authors elaborate on these criteria? Was only circular sequences included in the final comparison? What about presence of linear plasmids, which has at least been described in Salmonella and Klebsiella in the literature? Also, was circular sequences with more than one replicon included in the analysis? These could be miss-assemblies but may also be correct, as especially IncF and IncHI plasmids can carry more than one replicon.

Response: Multi-replicon plasmids were included in the analysis. Assemblies with non-circular contigs were excluded. We accept that linear plasmids have been rarely described in the literature and we may be excluding some of these in our analysis but they are difficult to distinguish from the more probable scenario of incomplete assemblies. We have now acknowledged this limitation in the discussion:

“We chose to exclude assemblies with non-circularised plasmids which may exclude linear plasmids, though these are difficult to differentiate from incomplete circular plasmids.”

E) What %ID cut-off was used for detecting replicons in PlasmidFinder and MOB genes in MOB-Suite? The choice would have had consequences for correct detection of plasmids as e.g. PlasmidFinder is designed to have a cut-off at 80% to detect similar (small) plasmids. Was this cut-off used in the initial analysis before the tool was discarded?

Response: PlasmidFinder was run using an 80% minimum ID cut-off, similarly mob-suite was run on default settings (minimum 80% relaxase ID). We apologise that this was not clear in the submission and have now included this detail in the methods:

“..we therefore used two established methods that have been applied to large-scale short read sequencing datasets, Replicon typing using PlasmidFinder (--minid 80) and Relaxase typing with MOB-suite (default settings i.e. --min_mob_cov 80).”

F) On what basis was plasmids categorized into small (>10kb), medium (>10kb; <100kb) and large plasmids (>100 kb)? These numbers seems rather arbitrary. It may have been more relevant to group according to other criteria than size. Or is the division into these three groups based on a size distribution of all the identified plasmids and thus represents the 3 most meaningful peaks in such a distribution analysis?

Response: Please see our response to Reviewer#1.

G) As neither replicon nor MOB typing was helpful to group plasmids, a new method described in the manuscript by the sentence; 'Subsequently, we therefore opted to use a previously described classification approach, utilizing a graph-based Louvain community detection algorithm(10)'. It is not clear from the Materials section, exactly how good this method was to group the different plasmids into meaningful groups/clusters. Please explain in more details, how closely related two plasmids should be in order to be clustered together? What if e.g. a 20% deletion occurred by a single genetic event in a plasmid. Would the original plasmid and the plasmid with the deletion both be put into the same clusters? How was this approach validated to ensure that the clustering made biological sense?

Response: As explained in the methods section we validated the clustering against existing replicon/relaxase typing methods using the normalised mutual information index. This demonstrated high congruence with these methods despite the fact that unlike other recent similar approaches e.g. <https://doi.org/10.1038/s41467-020-16282-w> we did not use these to derive the sparsifying threshold.

H) The findings concerning the plasmid clustering given in the manuscript is difficult to translate to existing plasmid typing nomenclature. To make the findings presented in the manuscript more applicable for others, some sort of connection between the (unnamed clusters) and replicon-and MOB-types should be considered.

Response: Thank you - we agree with this and have made references throughout the text between the plasmid groups described and replicon types/Plasmid taxonomic units (PTUs). We have also annotated Fig.4 to make this connection clearer to the reader.

Additional References

- 1 David S, Cohen V, Reuter S, *et al.* Integrated chromosomal and plasmid sequence analyses reveal diverse modes of carbapenemase gene spread among *Klebsiella pneumoniae*. *Proc Natl Acad Sci U S A* 2020; published online Sept 23. DOI:10.1073/pnas.2003407117.
- 2 Hawkey J, Wyres KL, Judd LM, *et al.* ESBL plasmids in *Klebsiella pneumoniae*: diversity, transmission and contribution to infection burden in the hospital setting. *Genome Med* 2022; **14**: 97.
- 3 Goswami C, Fox S, Holden MTG, Connor M, Leanord A, Evans TJ. Origin, maintenance and spread of antibiotic resistance genes within plasmids and chromosomes of bloodstream isolates of *Escherichia coli*. *Microb Genom* 2020; **6**. DOI:10.1099/mgen.0.000353.
- 4 Peter S, Bosio M, Gross C, *et al.* Tracking of Antibiotic Resistance Transfer and Rapid Plasmid Evolution in a Hospital Setting by Nanopore Sequencing. *mSphere* 2020; **5**. DOI:10.1128/mSphere.00525-20.
- 5 Zong Zhiyong, Partridge Sally R., Thomas Lee, Iredell Jonathan R. Dominance of blaCTX-

M within an Australian Extended-Spectrum β -Lactamase Gene Pool. *Antimicrob Agents Chemother* 2008; **52**: 4198–202.

- 6 Agyekum A, Fajardo-Lubián A, Ansong D, Partridge SR, Agbenyega T, Iredell JR. blaCTX-M-15 carried by IncF-type plasmids is the dominant ESBL gene in Escherichia coli and Klebsiella pneumoniae at a hospital in Ghana. *Diagn Microbiol Infect Dis* 2016; **84**: 328–33.
- 7 Partridge SR, Zong Z, Iredell JR. Recombination in IS26 and Tn2 in the evolution of multiresistance regions carrying blaCTX-M-15 on conjugative IncF plasmids from Escherichia coli. *Antimicrob Agents Chemother* 2011; **55**: 4971–8.
- 8 Partridge SR, Ellem JA, Tetu SG, Zong Z, Paulsen IT, Iredell JR. Complete sequence of pJIE143, a pir-type plasmid carrying ISEcp1-blaCTX-M-15 from an Escherichia coli ST131 isolate. *Antimicrob Agents Chemother* 2011; **55**: 5933–5.
- 9 Partridge SR. Analysis of antibiotic resistance regions in Gram-negative bacteria. *FEMS Microbiol Rev* 2011; **35**: 820–55.
- 10 Partridge SR, Kwong SM, Firth N, Jensen SO. Mobile Genetic Elements Associated with Antimicrobial Resistance. *Clin Microbiol Rev* 2018; **31**. DOI:10.1128/CMR.00088-17.
- 11 Lipworth SIW, Vihta KD, Chau K, *et al*. Ten year longitudinal molecular epidemiology study of Escherichia coli and Klebsiella species bloodstream infections in Oxfordshire, UK. *Genome Med* 2021.
- 12 Kallonen T, Brodrick HJ, Harris SR, *et al*. Systematic longitudinal survey of invasive Escherichia coli in England demonstrates a stable population structure only transiently disturbed by the emergence of ST131. *Genome Res* 2017; published online July 18. DOI:10.1101/gr.216606.116.
- 13 Gladstone RA, McNally A, Pöntinen AK, *et al*. Emergence and dissemination of antimicrobial resistance in Escherichia coli causing bloodstream infections in Norway in 2002–17: a nationwide, longitudinal, microbial population genomic study. *The Lancet Microbe* 2021; published online May 10. DOI:10.1016/S2666-5247(21)00031-8.
- 14 Wick RR, Judd LM, Gorrie CL, Holt KE. Unicycler: Resolving bacterial genome assemblies from short and long sequencing reads. *PLoS Comput Biol* 2017; **13**: e1005595.

Reviewer #1 (Remarks to the Author):

I previously reviewed the paper by Lipworth et al. After carefully reading the revised version of this work and the responses to the reviewer's comments I should acknowledge the efforts made by the authors. They significantly improved the manuscript and addressed most of the concerns raised by the referees. Of special value is the comparison between some methods used for the analysis of plasmidome when large datasets are used (COPLA, MOB-suite,..) and the technical information provided to use the tools. Below, I comment on two major aspects of the paper. Still, one point can be further clarified.

- Novelty of some statements. In my previous review, I criticized the novelty of some of the main messages (e.g. involvement of a few plasmid types in the carriage of ARGs) which were previously described in the literature. In response to these concerns, the authors properly and detailed argued how the sample and the goals of such publications differed from the present work. Of course, I fully agree with that. The high value of Lipworth's paper is the sample analyzed (longitudinal, local) which allows for analyzing the transmission pathways in a given region more accurately than cross-sectional studies (often including a limited sample) could provide. I do believe that the results confirm some "knowledge background" for what scientific evidence can result necessary by using high-resolution analysis. The authors highlight "this novelty" in the current version of the manuscript and supported some of the mentioned aspects with references. In addition, they further highlighted other novel and valuable aspects of the paper (the methodological part is of special value).
- Redundancy. A point that needs to be clarified is the presence of isolates from outbreaks which could result in the overrepresentation of certain PGs. This concern was raised by different referees but it is still not clear in the text. In the methods they said ("Additionally we selected isolates from clinically important local AMR-associated outbreaks and representatives of other species with apparently similar plasmidomes"). In the response to the referees, the authors mentioned that "isolates were selected in an unbiased and sequential manner". What does it mean? And they added "Whilst it is true that sequencing isolates from dominant clones can bias the data, it is also true that such clones dominate the population structure of Oxfordshire BSI isolates (and global isolates in the case of E. coli). Although I agree with the comment, I believe that a clear description in the material and methods section (the number of clonally related isolates) and a comment about this in the text are still.

Minor changes.

1. Change "prevelant" to "prevalent".
2. Revise italics in the name of the genes along the text.
3. Pag 5. Second column, 5th line from the bottom. What is transferred is a transposable unit containing blaCTXM15 and other genes, not single genes. Revise the sentence.

Reviewer #2 (Remarks to the Author):

[No further comments for author]

Reviewer #3 (Remarks to the Author):

The authors have done an excellent job incorporating feedback from each of the different reviewers and have presented a well thought out and clear manuscript. I have some minor suggestions for wording changes that they can chose to incorporate but are definitely not essential. I recommend for publication

- Assingation is correctly used but is not a commonly seen word, I recommend using assignment instead

- At various points the authors use "unselected" to refer to their sampling strategy and I appreciate that from the review process, the term "unbiased" is definitely a contentious word. Depending on the context of its use at the different points in the manuscript I recommend perhaps

using "inclusive" sampling to encapsulate the premise that you are sampling isolates not based strictly on a specific AMR profile

Reviewer #1 (Remarks to the Author):

I previously reviewed the paper by Lipworth et al. After carefully reading the revised version of this work and the responses to the reviewer's comments I should acknowledge the efforts made by the authors. They significantly improved the manuscript and addressed most of the concerns raised by the referees. Of special value is the comparison between some methods used for the analysis of plasmidome when large datasets are used (COPLA, MOB-suite,...) and the technical information provided to use the tools. Below, I comment on two major aspects of the paper. Still, one point can be further clarified.

- Novelty of some statements. In my previous review, I criticized the novelty of some of the main messages (e.g. involvement of a few plasmid types in the carriage of ARGs) which were previously described in the literature. In response to these concerns, the authors properly and detailed argued how the sample and the goals of such publications differed from the present work. Of course, I fully agree with that. The high value of Lipworth's paper is the sample analyzed (longitudinal, local) which allows for analyzing the transmission pathways in a given region more accurately than cross-sectional studies (often including a limited sample) could provide. I do believe that the results confirm some "knowledge background" for what scientific evidence can result necessary by using high-resolution analysis. The authors highlight "this novelty" in the current version of the manuscript and supported some of the mentioned aspects with references. In addition, they further highlighted other novel and valuable aspects of the paper (the methodological part is of special value).

Thankyou very much for the time you have taken to review our manuscript and we are glad that you found our response to be helpful.

- Redundancy. A point that needs to be clarified is the presence of isolates from outbreaks which could result in the overrepresentation of certain PGs. This concern was raised by different referees but it is still not clear in the text. In the methods they said ("Additionally we selected isolates from clinically important local AMR-associated outbreaks and representatives of other species with apparently similar plasmidomes"). In the response to the referees, the authors mentioned that "isolates were selected in an unbiased and sequential manner". What does it mean?

In our previous response we stated that "The breakdown of isolates is shown in figure S1 - 547/738 (74%) of isolates were selected in an unbiased and sequential manner."

E. coli and *Klebsiella* spp. isolates were selected for sequencing in a sequential, non-biased manner. The details of this are in the Isolate selection section in the methods "In this earlier study we sequenced all available isolates in this (2009-2018) time period (i.e. a non-biased, sequential and near complete dataset; n=3468 isolates) with de-duplication to 90 days per patient". As stated in Figure S1, this represents 74% of the sample sequenced in this study. We have added this detail to the methods section as follows:

“In the current study, we additionally sequenced all *E. coli* and *Klebsiella* spp. isolates from 2009 and 2018 using Oxford Nanopore Technologies (547/738, 74% of isolates successfully sequenced Figure S1).”

And they added “Whilst it is true that sequencing isolates from dominant clones can bias the data, it is also true that such clones dominate the population structure of Oxfordshire BSI isolates (and global isolates in the case of *E. coli*). Although I agree with the comment, I believe that a clear description in the material and methods section (the number of clonally related isolates) and a comment about this in the text are still.

We have added further details about the clonality of the enriched set as requested:

“Of the 191/738 (26%) isolates successfully sequenced as part of this enriched dataset, 17/191 (9%) were *K. pneumoniae* ST 490 (the dominant ST in Oxfordshire over this time period(12)) and 55/191 (29%) belonged to the predominant *E. coli* sequence types (STs 131/95/73/69).”

And we have added the relevant points from our previous rebuttal to the discussion:

“Whilst the decision to include an enriched sample (representing 26% of the total dataset in this study) can bias the data with over-representation of dominant clones, it is also true that such clones dominate the population structure of Oxfordshire BSI isolates (and global isolates in the case of *E. coli*(12, 18, 19)).”

Minor changes.

1. Change “prevelant” to “prevalent”.

Thanks for spotting, we have corrected.

2. Revise italics in the name of the genes along the text.

We have checked the manuscript to ensure all gene names are in italic but are happy to comply with any further editorial requests to meet the house style of Nature Communications.

3. Pag 5. Second column, 5th line from the bottom. What is transferred is a transposable unit containing blaCTXM15 and other genes, not single genes. Revise the sentence.

We have incorporated this suggestion as follows:

“Inspection of core-genome phylogenies of the two largest *bla*_{CTX-M-15} carrying plasmid groups...demonstrated multiple probable independent horizontal acquisition events of transposable units containing this gene”

Reviewer #2 (Remarks to the Author):

[No further comments for author]

Reviewer #3 (Remarks to the Author):

The authors have done an excellent job incorporating feedback from each of the different reviewers and have presented a well thought out and clear manuscript. I have some minor suggestions for wording changes that they can chose to incorporate but are definitely not essential. I recommend for publication

Thankyou for taking the time to review our manuscript. We are pleased that you found our revisions to be satisfactory.

- Assingation is correctly used but is not a commonly seen word, I recommend using assignment instead

Thankyou for this suggestion which we have incorporated.

- At various points the authors use “unselected” to refer to their sampling strategy and I appreciate that from the review process, the term “unbiased” is definitely a contentious word. Depending on the context of its use at the different points in the manuscript I recommend perhaps using “inclusive” sampling to encapsulate the premise that you are sampling isolates not based strictly on a specific AMR profile

Thanks for this suggestion, however we would prefer to continue to use the term “unselected”. We used the term “unselected” to indicate that isolates are not selected based on AMR profile or any other characteristic (e.g. we attempt to sequence all isolates collected over a given time period in a sequential manner). We feel that this distinction is important and is a different strategy compared to being inclusive of isolates regardless of their AMR profile.